# Interpretable point cloud classification using multiple instance learning

## Abstract

3D image analysis is crucial in fields such as autonomous driving and biomedical research. However, existing 3D point cloud classification models lack interpretability, limiting trust and usability in safety-critical applications. To address this, we propose POINTMIL, an inherently locally interpretable point cloud classifier using Multiple Instance Learning (MIL). POINTMIL offers local interpretability, providing fine-grained point-specific explanations to point-based models without the need for *post-hoc* methods, addressing the limitations of global or imprecise interpretability approaches. We applied POINTMIL to four popular point cloud classifiers, PointNet, DGCNN, CurveNet, PointMLP and PointNeXt, and proposed a transformer-based backbone to extract high-quality point-specific features. POINTMIL made these models inherently interpretable while increasing predictive performance on standard benchmarks (ModelNet40, ShapeNetPart) and achieving state-of-the-art mACC (97.3%) and F1 (97.5%) on the IntrA biomedical data set, and another dataset of biological cells. To our knowledge, this is the first work to apply MIL to interpretable point cloud classification.

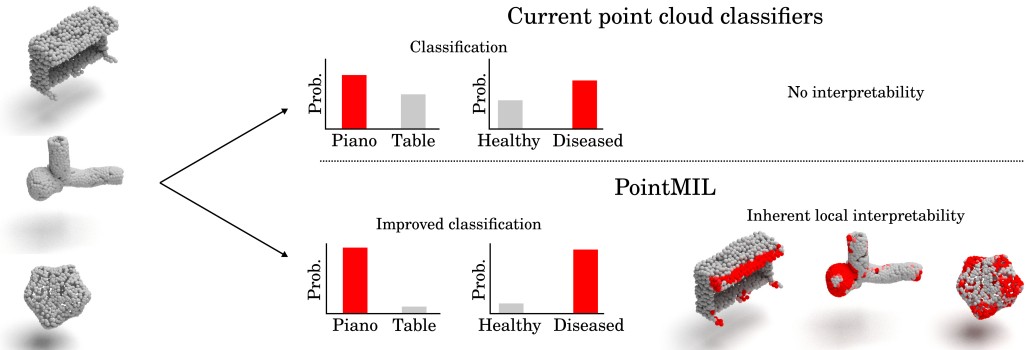

Figure 1: Current point cloud classifiers usually only provide predictive probabilities. We propose POINTMIL to inherently incorporate interpretability and improve predictive performance into point-based architectures.

## 1 Introduction

Three-dimensional (3D) imaging data is prevalent in various fields, including autonomous driving, augmented reality, robotics, and biology. In autonomous driving, 3D point clouds enable vehicles to perceive and navigate their surroundings safely, identifying obstacles and road features. In biology, the 3D shape of cells has provided insight into the underlying cell state (Viana et al., 2023), enabling advances in diagnostics (Song et al., 2024) and drug discovery.

Significant progress has been made in the processing of point clouds representations of 3D shapes for classification and segmentation tasks (Guo et al., 2020). However, most methods do not explain their decision-making, which limits adoption in real world scenarios due to concerns about safety and trustworthiness (Rudin, 2019; Rudin et al., 2022). Despite significant advancements in the

interpretability of machine learning models in 2D image analysis (Zhang et al., 2021; Wang et al., 2023; Hu et al., 2024; Paul et al., 2024), there has been a lack of research on the interpretability of 3D point cloud models. More so, of those proposed, the majority are either *post-hoc*, meaning that an extra modelling step is required to obtain interpretations, or they are *globally* interpretable, meaning that they lack the ability to offer fine-grained, point-specific explanations.

To address these challenges and elucidate the model's decision-making process, we propose POINT-MIL, an inherently interpretable classification framework for point clouds that offers fine-grained, *local* and class-specific interpretations using Multiple Instance Learning (MIL; Dietterich et al. (1997)). Given its ability to handle data organised into bags of instances, MIL is well suited for point cloud analysis, especially in bioimaging domains, where each point in a point cloud is assigned the same label, but only certain points are discriminatory (Yang et al., 2020). Building on this foundation, we present a model that leverages the strengths of MIL to offer robust performance and interpretability in point cloud classification. Furthermore, we introduce a contextual attention mechanism, which incorporates neighbourhood information into the attention calculation, addressing the sparsity of traditional attention methods and enabling smoother, more coherent attention distributions. This adaptation ensures that the model can better capture local geometric relationships within the point cloud, improving both classification performance and interpretability. Our main contributions are as follows:

1. We propose POINTMIL, a point-based classification pipeline based on MIL, to offer inherent *local* interpretability and enhanced classification performance to existing point-based feature extractors.

2. We adapt and introduce a new transformer-based model to extract high-quality point-specific features from a point cloud.

3. We incorporate contextual attention to address sparsity in attention weights, improving interpretability and classification performance by leveraging local neighbourhood information.

4. We show the generality of POINTMIL on *de-facto* public benchmarks (ModelNet40 (Wu et al., 2015) and ShapeNetPart (Yi et al., 2016)) and biomedical imaging datasets, achieving the state-of-the-art (SOTA) on IntrA (Yang et al., 2020).

## 2   RELATED WORK

**Point cloud analysis:** One of the first methods that used unordered point clouds directly for classification and segmentation was PointNet (Qi et al., 2017a). PointNet, however, ignored local relationships between points. Subsequently, PointNet++ (Qi et al., 2017b) introduced hierarchical feature learning to capture locality recursively. Many modern algorithms are built on the design philosophy of PointNet++, including convolutional kernel-based (Li et al., 2018b; Thomas et al., 2019; Wu et al., 2019), graph-based (Wang et al., 2019a;b; Xu et al., 2020), MLP-based (Choe et al., 2022; Ma et al., 2022), and transformer-based methods (Zhang et al., 2020; Zhao et al., 2021; Guo et al., 2021; Yu et al., 2021; Cheng et al., 2022; Akwensi et al., 2024). Although significant progress has been made in advancing classification and segmentation accuracy, little work has focused on interpretability.

**Interpretability on point clouds:** Interpretability methods can be classified along two key dimensions: (1) the stage at which interpretability is introduced and (2) the scope of the explanations provided. Regarding the stage, methods are either *post-hoc* or *inherently interpretable*. *Post-hoc* methods generate explanations after the model has made its predictions, often through additional analysis, approximation techniques, or assessing gradients with respect to the input (Zhou et al., 2016). In contrast, *inherently interpretable* methods are designed to integrate interpretability into the model itself, producing explanations as part of the prediction process. With respect to scope, methods are categorised as either *local* or *global*. *Local* approaches focus on explaining individual predictions, offering insights specific to a single input. *Global* approaches aim to provide a holistic understanding of the model's behaviour across all inputs. Since PointNet ++ (Qi et al., 2017b), many point-based models have used some form of sampling and grouping (Guo et al., 2021; Zhao et al., 2021; Xiang et al., 2021; Ma et al., 2022), thus losing point-level information in the classification stage. Therefore, most *local* interpretability methods for point cloud classification are *post-hoc*, including gradient-based (Zhang et al., 2019; Huang et al., 2020) and surrogate models (Tan & Kot-

thaus, 2022) based on LIME (Ribeiro et al., 2016). Zhang et al. (2019) and Huang et al. (2020) developed explainability methods for PointNet using global average pooling (GAP) and class activation maps. Taghanaki et al. (2020) introduced a module into point set encoders that masked points with negligible contributions, leaving only informative points in the classification layer. Similarly, Zheng et al. (2019) obtained saliency maps by shifting points to the object centroid and calculating the corresponding loss gradient with respect to the shifted points. However, *post hoc* methods have been shown to be deceptive and often troublesome (Laugel et al., 2019; Rudin et al., 2021; Feng et al., 2024). For example, the interpretations of *post hoc* methods can differ depending on the interpretability methods (Li et al., 2018a), leading to convincing but conflicting interpretations for the same classification. *Post-hoc* methods also involve an additional modelling step, raising further concerns about the precision of their interpretations Fan et al. (2021). Few inherently interpretable methods for point cloud classifications have been proposed, and of these, most are *global*. Arnold et al. (2023) developed XPCC, a prototype-based interpretable model that used point cloud representation distributions to learn class-specific prototypes. Similarly, Feng et al. (2024) developed Interpretable3D, a prototype-based global interpretability model that can be used in conjunction with other model architectures for classification and segmentation. However, none of these inherently interpretable methods offers local interpretations on a point-level basis. While global interpretability provides valuable insights into the overall behaviour of a model, local methods can be especially beneficial when understanding specific, individual predictions is crucial, offering more granular and context-sensitive explanations. To our knowledge, no one has yet offered an inherently *locally* interpretable model for point cloud classification. POINTMIL utilises MIL to offer an inherently *locally* interpretable model.

**Multiple instance learning:** In the typical binary MIL problem, a bag is labelled positive if and only if at least one of its instances is labelled positive (Dietterich et al., 1997); however, there is no access to individual instances during training. MIL algorithms then attempt to classify entire bags of instances and often pinpoint important or class conditional discriminatory instances as interpretability output. Many MIL methods have been proposed for drug activity prediction (Dietterich et al., 1997), video image analysis (Ali & Shah, 2010), and cancer detection and sub-typing (Ilse et al., 2018; Shao et al., 2021; Lu et al., 2021; Fourkioti et al., 2024). Recently, Early et al. (2024) extended MIL to time series classification in an interpretable plug-and-play framework. However, to our knowledge, no one has used MIL for interpretable point cloud classification.

## 3 METHODS

Given a point cloud $\mathbf{P} \in \mathbb{R}^{N \times 3} = \{\mathbf{p}_i | i = 1, \ldots, N\}$, consisting of $N$ points in Cartesian space $(x, y, z)$, and their associated $d$-dimensional point features (often point normals, however, these can be the point coordinates if no point-level features exist) $\mathbf{F} \in \mathbb{R}^{N \times d_{in}} = \{\mathbf{f}_i | i = 1, \ldots, N\}$, traditional point-based methods use a point-based encoder $f_{enc}$ to learn a global representation $\mathbf{z} \in \mathbb{R}^d$ for $\mathbf{P}$ by aggregating the points with equal weighting (often through adaptive pooling), followed by a classification head $f_{clf}$.

We propose a new approach by learning a representation $\mathbf{z}_i \in \mathbb{R}^d$ for each point $\mathbf{p}_i$ for $i \in \{1, \ldots, N\}$, and then applying MIL pooling for simultaneous classification and interpretability. Our framework consists of a point-based feature extractor $f_{enc}$ and a MIL pooling module $f_{MIL}$.

### 3.1 FEATURE EXTRACTOR

To develop a point-level feature extractor, we follow much of the Transformer block from Yu et al. (2021). However, unlike Yu et al. (2021), we did not use point sampling strategies. Furthermore, we did not use their multi-graph reasoning. This feature extractor aimed to incorporate contextual information into the point cloud features by: (1) grouping points with $k$-Nearest Neighbours ($k$-NN), (2) including relative positional embeddings, and (3) refining point-level features through an attention mechanism. These are detailed in Appendix A.

We also presented analysis on PointNet (Qi et al., 2017a), DGCNN (Wang et al., 2019b), CurveNet (Xiang et al., 2021), PointMLP (Ma et al., 2022), and PointNeXt (Qian et al., 2022) feature extractors. For PointNet and DGCNN we replaced the classification heads of these architectures with MIL pooling described in Section 3.2. CurveNet and PointMLP downsample the original point cloud. In

order to retain point-level features for every point, we slightly adapted these architectures to remove point sampling. We show the affect of this adaptation on classification results so that any difference in performance can then be attributed to the MIL pooling instead of this adaptation. We used PointMLPElite for our analysis. For PointNeXt-S, we slightly adapted the architecture such that point-level features from the first layer were concatenated with global features from the last layer before input into our MIL pooling. These adaptations are discussed further in Appendix A. Each feature extractor produced $d$-dimensional point-level features $\mathbf{Z} \in \mathbb{R}^{N \times d} = f_{enc}(\mathbf{P})$, for $N$ points which were fed into different MIL pooling algorithms.

## 3.2 MIL POOLING

After obtaining feature representations $\mathbf{z}_i$ for each point $\mathbf{p}_i$, we evaluated four MIL pooling methods that offer inherent interpretability, `Instance` (Wang et al., 2018), `Attention` (Ilse et al., 2018), `Additive` (Javed et al., 2022), and `Conjunctive` (Early et al., 2024).

`Instance` pooling predicts the label of each point through an instance classifier and then pools the predictions by taking the mean:

$$\hat{\mathbf{y}}_i \in \mathbb{R}^c = f_{clf}(\mathbf{z}_i); \qquad \hat{\mathbf{Y}} = \frac{1}{N} \sum_{i=1}^{N} (\hat{\mathbf{y}}_i), \tag{1}$$

where $c$ is the number of classes.

`Attention` pooling calculates the attention weights of the point features through an MLP, calculates a weighted average feature representation for the point cloud using those weights and then classifies that features using an MLP:

$$a_i \in [0, 1] = f_{attn}(\mathbf{z}_i); \qquad \hat{\mathbf{Y}} = f_{clf}\left(\frac{1}{N} \sum_{i=1}^{N} a_i \mathbf{z}_i\right). \tag{2}$$

`Additive` pooling calculates attention weights for each point feature, then classifies each point according to its weighted feature vector, and finally produces a bag classification from the mean of all weighted instance classifications:

$$a_i \in [0, 1] = f_{attn}(\mathbf{z}_i); \quad \hat{\mathbf{y}}_i = f_{clf}(a_i \mathbf{z}_i); \quad \hat{\mathbf{Y}} = \frac{1}{N} \sum_{i=1}^{N} (\hat{\mathbf{y}}_i). \tag{3}$$

`Conjunctive` pooling trains the point attention and point classification heads independently so that attention weights and point predictions are computed on the features alone. The final point cloud classification is given by the weighted sum of the point classifications weighted by the attention weights:

$$a_i \in [0, 1] = f_{attn}(\mathbf{z}_i); \quad \hat{\mathbf{y}}_i = f_{clf}(\mathbf{z}_i); \quad \hat{\mathbf{Y}} = \frac{1}{N} \sum_{i=1}^{N} (a_i \hat{\mathbf{y}}_i). \tag{4}$$

## 3.3 CONTEXTUAL ATTENTION

As Early et al. (2024) showed that these pooling operations often produced sparse explanations which occasionally did not cover the entire discriminatory regions, we propose injecting a contextual prior into our calculation of attention, following ideas similar to Fourkioti et al. (2024). For attention-based pooling methods, `Attention`, `Additive`, and `Conjunctive`, attention weights for each point are calculated as:

$$a_i \in [0, 1] = f_{attn}(\mathbf{z}_i), \tag{5}$$

where $f_{attn}$ is an MLP and $\mathbf{z}_i$ is a feature vector for each point $\mathbf{p}_i$. We propose updating these attention weights according to the attention weights of the nearest neighbours of each point $i$, such that:

$$a_i^{\text{new}} \in [0, 1] = \frac{1}{k} \sum_{j \in \mathcal{N}(\mathbf{p}_i)} a_j, \tag{6}$$

Table 1: Interpretability results in terms of AOPCR and NDCG@n (AOPCR/NDCG@n) on IntrA. The best results are given for each method in **bold**.

|  | PointNet | DGCNN | CurveNet | PointNeXt | Transformer |
|---|---|---|---|---|---|
| PSM | 0.579/0.243 | 0.916/0.248 | 1.371/0.218 | 0.092/0.272 | 6.518/0.320 |
| CLAIM | 0.967/0.187 | **6.033**/0.480 | 1.363/0.252 | 0.226/0.294 | 14.023/0.593 |
| Add. | 0.792/**0.254** | 4.486/**0.482** | 0.615/**0.266** | 1.259/0.300 | **18.162**/**0.613** |
| Att. | 0.005/0.222 | −0.031/0.223 | 1.520/0.260 | 0.044/0.235 | 14.541/0.539 |
| Conj. | 0.741/0.208 | 4.828/0.467 | **2.660**/0.207 | 1.531/**0.310** | 16.305/0.610 |
| Inst. | **0.973**/0.225 | 5.212/0.462 | 1.709/0.236 | **2.160**/0.285 | 16.166/0.587 |

where $\mathcal{N}(\mathbf{p}_i)$ represents the set of points in the neighbourhood of $\mathbf{p}_i$. This update mechanism smooths the attention weights by incorporating the information from the local neighbourhood, thus addressing the sparsity of the original attention mechanism and providing a more context-aware attention distribution across the point cloud.

## 3.4 INTERPRETABILITY

Interpretations were derived through MIL pooling. The `Instance` pooling strategy classifies each point individually before pooling, yielding point-level predictions: $\{\hat{\mathbf{y}}_i | i = 1, \ldots, N\}$. `Additive` and `Conjunctive` also make point-level predictions; however, the interpretations are scaled by attention weights: $\{a_i \hat{\mathbf{y}}_i | i = 1, \ldots, N\}$. For each of these pooling algorithms, we applied a softmax operation over the class dimension and took the index of the class for which we wished to obtain interpretations, so that we obtained a scalar for each point in the point cloud. For the `Attention` pooling strategy, we used the attention weights: $\mathbf{a} \in \mathbb{R}^{1 \times N} = \{a_i | i = 1, \ldots, N\}$, which were interpreted as a measure of general importance for each point in the point cloud and were not class-specific.

## 4 EXPERIMENTS

We compared the interpretability of POINTMIL with other *locally* interpretable point cloud classification methods including class attentive interpretable mapping (CLAIM; Huang et al. (2020)), and point cloud saliency maps (PSM; Zheng et al. (2019)). Similarly to class activation maps (CAM; Zhou et al. (2016)), CLAIM uses global average pooling (GAP) after point-level feature extractors (the original paper focused on PointNet) and projects the weights of the classifier after GAP on the features of each point to obtain interpretations for each point. PSM assigns scores to each point based on its contribution to the classification loss. This is done by shifting the points towards the centroid of the point cloud and then calculating the gradient of the loss with respect to each point

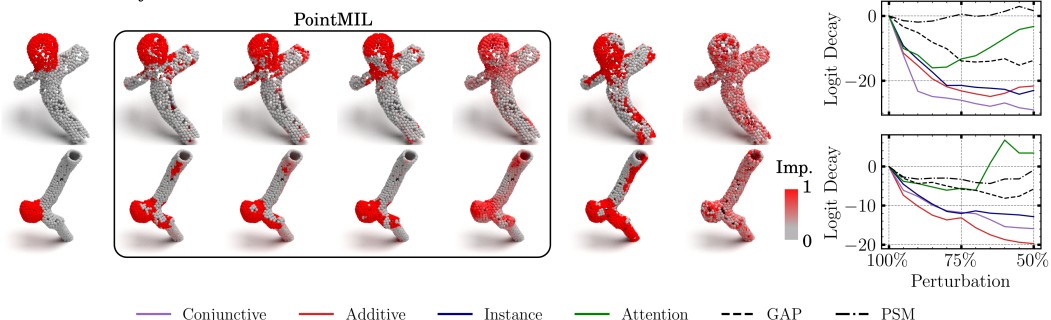

Figure 2: POINTMIL, CLAIM and PSM interpretability visualisations and corresponding perturbation curves using the Transformer backbonfor example cells from the IntrA dataset.

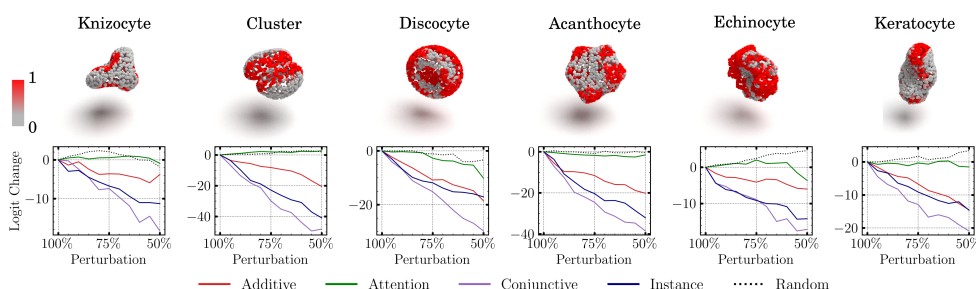

Figure 3: Interpretability visualisation (top row) and corresponding perturbation (bottom row) curves for different RBC shapes.

in spherical coordinates. We then compared POINTMIL to several other point-based architectures in terms of classification performance and assessed how the MIL pooling affected the results of the original backbones in segmentation tasks.

## 4.1 EVALUATION METRICS

We used the area over the perturbation curve to random (AOPCR; Samek et al. (2017)) and normalised discounted cumulative gain at $n$ (NDCG@n) to quantitatively evaluate interpretability (Early et al., 2022; 2024). Please see Appendix B for more details. For classification, we used the overall accuracy (oACC), mean class accuracy per class (mACC), and the F1 score. For segmentation, we used the average class intersection of union (IoU) and the instance IoU.

## 4.2 DATASETS

We evaluated POINTMIL on several open source datasets, including two real-world datasets of 3D cell shapes (IntrA (Yang et al., 2020) and 3D red blood cell (RBC) dataset (Simionato et al., 2021)) and two of everyday objects (ModelNet40 (Wu et al., 2015) and ShapeNetPart (Yi et al., 2016)). See Appendix C for more details.

## 5 RESULTS

## 5.1 INTERPRETABILITY

Table 1 shows the interpretability results on the IntrA dataset for PointNet, DGCNN, CurveNet, PointNeXt and the Transformer backbone. POINTMIL provided better interpretability performance than both PSM and CLAIM, overall. Across backbones, POINTMIL had the highest AOPCR and NDCG@n. The only exception was CLAIM that had a higher AOPCR for the DGCNN backbone. Among the interpretability methods, the Transformer produced the highest AOPCR and NDCG@n results. This could be due to the attention mechanisms within the Transformer block that already enabled the model to focus on informative points, which is further exacerbated by the MIL pooling. Among all backbones, PointNet performed the worst, suggesting that PointNet is not adequate in

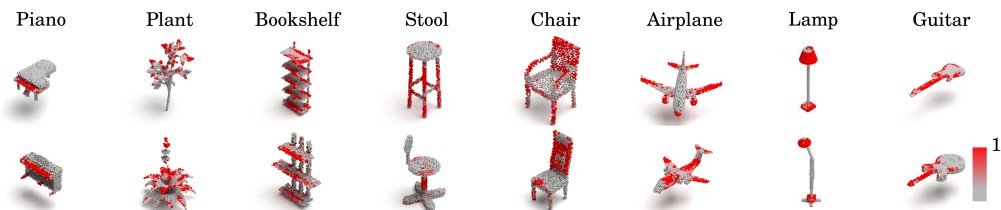

Figure 4: Interpretability outputs of `PointMIL` for different shape classes from ModelNet40

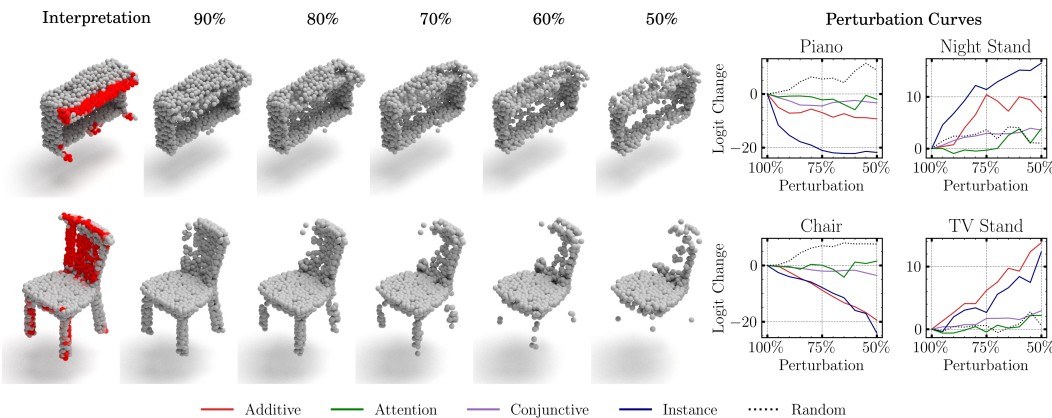

Figure 5: Interpretability outputs and perturbation curves of POINTMIL with the Transformer backbone for different shape classes from ModelNet40

capturing discriminative morphological cues. For PointNeXt, although the PointMIL versions outperformed PSM and CLAIM, the lower values when compared to DGCNN and the Transformer could be attributed to the concatenation of local with global features before the MIL pooling.

Visualisations of the interpretability for each pooling method on the annotated *Aneurysm* class using the Transformer backbone are shown in Figure 2. The red points indicate areas deemed significant by the model for that specific class. *Aneurysm's* are presented by the abnormal bulging or ballooning of blood vessels. The first column in Figure 2 shows local annotations of *Aneurysms*, with each other column presenting interpretations for the *Aneurysm*

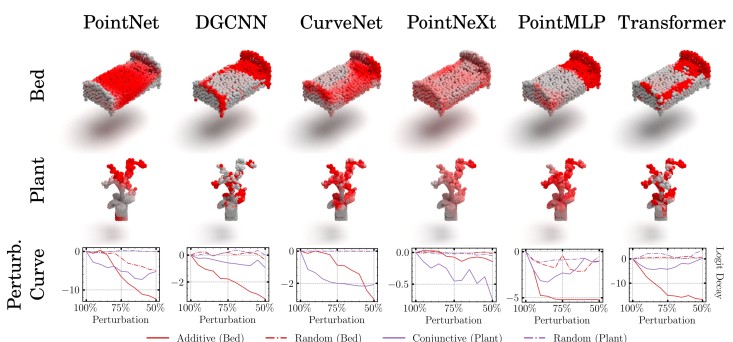

Figure 6: Interpretability of POINTMIL with different backbones on an example *Bed* (top row) and *Plant* (midle row) from ModelNet40. Perturbation curves are shown in the bottom row.

class using the different methods. The last columns show the perturbation curves. These show the decay in the logit of the predicted class after removing the most important points. A larger decay suggests that those points are indeed discriminative for the class. POINTMIL is clearly able to localise on informative regions better than other methods as seen by the visualisation as well as a larger decay in logits shown by the perturbation curve.

Among all MIL pooling methods, `Additive` and `Conjunctive` performed best on the IntrA dataset. This superior performance of `Additive` and `Conjunctive` pooling can be attributed to their ability to better aggregate point-level importance scores. Additive pooling scales point features with their importance weights, preserving detailed information while focusing on relevant points before being passed into a point-level classifier. Conjunctive pooling further enhances this by independently computing attention weights and class-specific contributions, explicitly aligning the model's focus with the predicted class. In contrast, `Instance` pooling lacks this importance weighting, and `Attention` pooling does not offer class-specific explanations and rather provides a general measure of importance across classes, which limits their interpretability.

We also present local interpretations for other datasets lacking ground truth annotations. Figure 3 illustrates the visual interpretations of POINTMIL with the Transformer backone for

six of the nine classes of RBC with their corresponding perturbation curves. This demonstrates that POINTMIL successfully localises on biologically relevant structural areas. For example, *Discocytes* are characterised by their biconcave shapes, with interpretations for this class focussing on regions identified around the central concavity. In the case of *Acanthocytes*, which exhibit several spicules of varying sizes that project from their surfaces at irregular intervals, POINTMIL similarly focused on these projections for identifying this class. For *Knizocytes*, which have a triangular morphology, the model highlighted the areas where the lobes converge. Additionally, POINTMIL pinpointed the spiky projections of *Echinocytes* and *Keratocytes*, as well as the interaction zones where two cells meet in *Cell Clusters*.

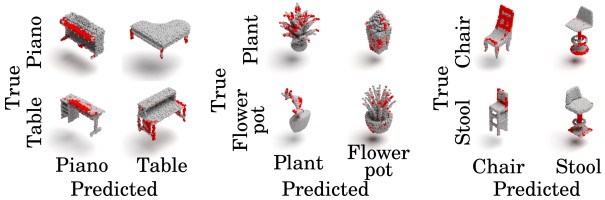

Figure 7: Interpretability visualisations of incorrect classifications from POINTMIL with Transformer backbone on ModelNet40.

POINTMIL is a versatile tool that is not limited to specific domains, making it suitable for a wide range of 3D shape classification tasks. Figure 4 presents the visual interpretations of POINTMIL applied to the ModelNet40 dataset, showcasing a subset of classes. For instance, when classifying a *Piano*, the model focused primarily on the keys, while the it emphasised on the branches and foliage of a *Plant*. The *Bookshelf* displayed red points along the shelves. Similarly, for the *Chair*, crucial features included the seat and legs, while the wings and fuselage were highlighted for *Airplane*. More examples are given in Appendix E.

Figure 5 shows the effect of removing the top 10% to 50% of important points on a *Piano* and *Chair* on the logits of those classes. The perturbation curves illustrate that when the points identified as most informative for classifying a *Piano* are removed, POINTMIL misclassifies the object as a Night Stand. Similarly, when the points identified as the most informative for classifying a *Chair* are removed, POINTMIL misclassifies the object as a TV stand. These interpretations reveal how POINTMIL effectively identified and localised relevant features across various object categories, enhancing our understanding of the model's decision-making process. Figure 6 presents the interpretability results for different backbones when classifying a *Bed* with `Additive` pooling (top row) and a *Plant* with `Conjunctive` pooling (middle row) from the ModelNet40 dataset. The perturbation curves are shown in the bottom row. Interestingly, DGCNN, CurveNet, PointMLP, and Transformer backbones consistently highlight similar regions of importance on the *Bed*, particularly focusing on the frame and headboard of the bed, which are key features distinguishing it from other objects. All backbones focussed on the leaves in the *Plant* as opposed to the pot. This consistency across backbones demonstrates the robustness of POINTMIL in identifying informative regions. Additionally, the agreement among backbones suggests that POINTMIL effectively leverages the feature representations generated by each model, ensuring the interpretability results are meaningful and aligned with the task. Finally, we demonstrated how POINTMIL could be used to assess where the model went wrong. For example, Figure 7 shows example confusion plots in which the attention of the predicted class is shown in red. Interestingly, for classifying plants, the model only focused on the plant, although when classifying flower pots, the model focused on both the flower and the pot.

## 5.2 CLASSIFICATION

Interpretability should promote classification accuracy and not hinder it. To showcase this, we performed classification on three separate datasets, two 3D biological cell-shape datasets, IntrA , and RBC, and the 3D shape classification benchmark ModelNet40. The results are shown in Table 2. POINTMIL outperformed all methods on IntrA and RBC in terms of mACC and F1 score by a considerable margin of at least $4.5\%$ and $3.3\%$ respectively. POINTMIL achieved SOTA on IntrA with an mACC of $97.3\%$ and an F1 of $97.5\%$ using `Conjunctive` pooling with the Transformer backbone. Importantly, POINTMIL increased the performance of all backbones on all datasets by up to $11.3\%$ in terms of mACC on RBC (shown in violet in Table 2). While POINTMIL was outperformed by recent SOTA methods like PointMLP (Ma et al., 2022), the original CurveNet (Xiang

Table 2: Classification results on IntrA, RBC, and ModelNet40. All results are shown without voting strategy on 1024 points. The highest results are shown in **bold**. Differences between backbones and POINTMIL are shown in violet. Adapted architectures without farthest point sampling results are shown with a †.

| Method | IntrA | | RBC | | ModelNet40 | |
|---|---|---|---|---|---|---|
| | mACC(↑) | F1(↑) | mACC(↑) | F1(↑) | mACC (↑) | oACC(↑) |
| PointNet(Qi et al., 2017a) | 81.8 | 82.4 | 67.7 | 67.1 | 86.2 | 89.2 |
| PointNet++(Qi et al., 2017b) | 92.7 | 94.2 | 86.2 | 87.1 | - | 91.9 |
| PointConv(Wu et al., 2019) | 83.0 | 82.1 | 68.1 | 67.9 | - | 92.5 |
| DGCNN Wang et al. (2019b) | 90.6 | 91.8 | 84.8 | 85.1 | 90.2 | 92.9 |
| PCT(Guo et al., 2021) | 69.2 | 68.9 | 68.7 | 69.2 | - | 93.2 |
| CurveNet(Xiang et al., 2021) | 88.3 | 89.8 | 88.3 | 87.8 | - | 93.8 |
| CurveNet† | 87.8 | 87.8 | 85.8 | 85.7 | 90.6 | 93.4 |
| PointMLP(Ma et al., 2022) | 88.4 | 88.8 | 91.8 | **92.2** | **91.3** | **94.1** |
| PointMLPElite | - | - | - | - | 90.9 | 93.6 |
| PointMLPElite† | - | - | - | - | 90.1 | 92.6 |
| PointNeXt(Qian et al., 2022) | 91.8 | 94.7 | 86.1 | 87.1 | 90.8 | 93.2 |
| 3DMedPT(Yu et al., 2021) | 92.2 | 93.3 | 81.3 | 83.2 | - | 93.4 |
| POINTMIL(PointNet) | 82.0+0.2 | 82.4+0.0 | 69.0+1.3 | 69.1+2.0 | 87.1+0.9 | 90.7+1.5 |
| POINTMIL(DGCNN) | 95.2 +3.2 | 94.6+2.8 | 92.4+7.6 | 92.4+7.3 | 90.8+0.6 | 93.1+0.2 |
| POINTMIL(CurveNet†) | 91.3+3.5 | 89.9+2.1 | 91.2+5.4 | 90.5+4.8 | 91.0+0.4 | 93.5+0.1 |
| POINTMIL(PointMLPElite†) | - | - | - | - | 90.5+0.4 | 93.5+0.9 |
| POINTMIL(PointNeXt) | 94.6+2.8 | 96.2+1.5 | 87.6+1.5 | 88.2+0.4 | 90.5−0.3 | 93.3+0.1 |
| POINTMIL(Trans.) | **97.3**+5.1 | **97.5**+4.2 | **92.6** +11.3 | **92.2**+9.0 | 89.0 | 92.7−0.7 |

et al., 2021) and PCT (Guo et al., 2021) on Modelnet40, POINTMIL outperformed these methods by considerable margins on IntrA and RBC. POINTMIL offered interpretability without harming and often improving classification performance.

## 5.3 ABLATION STUDIES

We evaluated the effect of including contextual attention in our attention-based pooling mechanisms: `Additive`, `Attention`, and `Conjunctive` and the impact of varying the value of $k$ (Figure 8). A value of $k = 0$ represented no contextual attention. Including contextual attention consistently offered advantages across all pooling methods and metrics compared to not using it. In terms of F1 and mACC contextual attention led to improved performance, particularly with the `Conjunctive` and `Attention` mechanisms, which consistently outperformed the `Additive` method as $k$ increased. All pooling methods produced F1 and mACC scores of $> 97\%$ after including contextual attention. For AOPCR, contextual attention was found to be most beneficial when using a value of $k = 12$. Lastly, considering NDCG@n, increasing $k$ provided the most benefit to `Attention` pooling, while offering slight improvements to `Additive` and `Conjunctive`. `Additive` and `Conjunctive` pooling outperformed `Attention` pooling across interpretability metrics, whether or not contextual attention was used. Although contextual pooling improved classification and interpretation methods, there is a trade-off in computation since the time complexity for

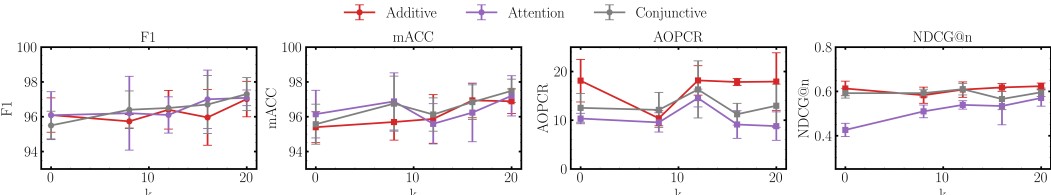

Figure 8: Ablation studies on the value of $k$ in our contextual attention on F1, mACC, AOPCR, and NDCG@n using the transformer backbone.

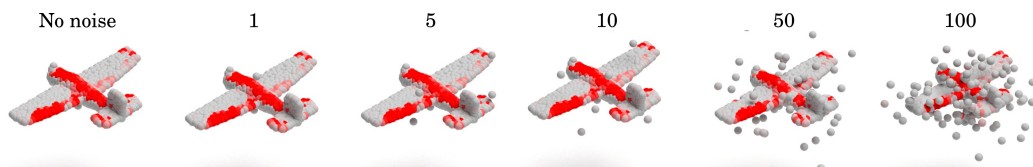

No noise      1      5      10      50      100

Figure 9: Interpretability visualisations of POINTMIL on a *Airlane* from ModelNet40 after adding a number (shown on the heading) of noisy points. POINTMIL is able to still focus on salient shape motifs ignoring noise.

$k$-NN graph search is $O(N^2)$ for the $N$ number of points. The graph construction time complexity is also $O(Nk)$, therefore, as $k$ increases, this process takes longer. We additionally demonstrate POINTMIL's robustness to noise. Figure 9 shows how, even when noisy points are added to objects, POINTMIL is still able to focus on salient 3D shape motifs. Further analysis is shown in Appendix F

### 5.4 SEGMENTATION

We evaluated POINTMIL for part segmentation on IntrA and ShapeNetPart using three of the five backbones. For IntrA, only the *Aneurysm* class contains annotations, therefore, we only reported metrics on this class. We followed the same settings as from Qi et al. (2017a) for segmentation on ShapeNetPart. The class-specific point-level interpretations were used as segmentation predictions. We assessed the `Conjunctive` and `Additive` MIL pooling as `Instance` was the equivalent to the original model's segmentation algorithms and `Attention` does not produce class-specific point-level classi-

Table 3: Segmentation results on IntrA and ShapeNetPart in terms of Class (Cls.) and Instance (Inst.) mIoU. The highest metrics are shown in **bold**.

| Method | IntrA IoU(↑) | ShapeNetPart Cls. IoU(↑) | ShapeNetPart Inst. IoU(↑) |
|---|---|---|---|
| PointNet | 72.2 | 81.7 | 84.2 |
| DGCNN | 76.4 | 83.6 | 85.2 |
| 3DMedPT | 82.4 | **84.3** | - |
| POINTMIL(PointNet) | 72.3 | 81.5 | 84.0 |
| POINTMIL(DGCNN) | 79.7 | 84.2 | **85.6** |
| POINTMIL(Trans) | **84.0** | 82.0 | 82.1 |

fication as interpretations. Interestingly, the segmentation results did not deteriorate and sometimes improved when using POINTMIL on both datasets. The only exception was 3DMedPT on ShapeNetPart, where the original 3DMedPT outperformed POINTMIL with the transformer backbone by a relatively larger margin.

## 6 CONCLUSION

In this work, we introduced POINTMIL, the first framework to apply MIL to point cloud classification, providing fine-grained point-specific interpretability without *post-hoc* techniques. We also introduced a contextual attention mechanism to adapt attention-based MIL to point clouds, accounting for the spatial and structural relationships inherent in 3D data. Using MIL, our approach improved both interpretability and classification performance on multiple backbones and datasets. POINTMIL achieved SOTA F1 and mACC by a significant margin. Future work could extend POINTMIL to consider using segmentation versions of other point-based models as backbones, as they provide point-specific features. Furthermore, analysis on more datasets that include point-specific ground-truth interpretation would help to better evaluate interpretability. The choice of pooling method should be guided by the specific requirements of the task and dataset characteristics. For tasks prioritising interpretability, `Conjunctive` pooling with contextual attention is recommended due to its class-specific focus. For applications prioritising simplicity, Instance pooling offers computational efficiency. An exploration of MIL pooling techniques specific to point cloud data could also enhance this work further. In conclusion, POINTMIL is a novel approach that effectively improved classification performance while providing inherent local interpretability, making it a valuable tool for 3D point cloud analysis in real-world applications.

## Reproducibility Statement

The code for this work was implemented in Python 3.10, with PyTorch and Lightning as the main machine learning libraries. The anonymous code is available at `https://anonymous.4open.science/r/PointMIL_ICLR-98B2/`. Model training was performed using an NVIDIA Tesla V100 GPU with 32GB of VRAM and CUDA v12.0 to enable GPU support.

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

## A    MODEL DETAILS

### A.1    TRANSFORMER BLOCK FEATURE EXTRACTOR

#### A.1.1    GROUP FEATURES THROUGH $k$-NEAREST NEIGHBOURS:

Formally, we constructed a $k$-NN graph on $\mathbf{P}$ with the graph including a self-loop to point-level features:

$$\mathcal{N}(\mathbf{p}_i) = \text{KNN}(\mathbf{P}, ||\mathbf{p}_i - \mathbf{p}_j||_2^2), \mathbf{p}_i, \mathbf{p}_j \in \mathbf{P},$$
$$\mathbf{f}_i' = [(\mathbf{f}_j - \mathbf{f}_i), \mathbf{f}_i]_{j \in \mathcal{N}(\mathbf{p}_i)} \in \mathbb{R}^{k \times 2d_{in}}, \tag{7}$$

where $\text{KNN}(\cdot)$ is the $k$-NN function, $[\cdot, \cdot]$ is concatenation, $k$ is the hyperparameter of the $k$-NN graph, $\mathcal{N}(\mathbf{p}_i)$ is the set of neighbours of $\mathbf{p}_i$, and $\mathbf{f}_i'$ is the point feature augmented with local contextual information.

#### A.1.2    LEARNED RELATIVE POSITIONAL ENCODING:

To encode spatial configurations per point-cloud neighbourhood we incorporated positional embeddings, $\mathbf{h}_i$ such that:

$$\mathbf{h}_i \in \mathbb{R}^{k \times d_h} = \phi_{pos}([\mathbf{p}_i - \mathbf{p}_j]_{j \in \mathcal{N}(\mathbf{p}_i)}), \tag{8}$$

where $\phi_{pos}$ is an MLP and $d_h$ is the output channel dimension of $\phi_{pos}$. The features were then further augmented with this positional encoding to give:

$$\mathbf{f}_i'' = [\mathbf{f}_i', \mathbf{h}_i]. \tag{9}$$

Thus, we obtained a new feature set $\mathbf{F}'' \in \mathbb{R}^{N \times k \times (2d_{in} + d_h)} = \{\mathbf{f}_i''\}_{i=1}^N$. This is then passed

#### A.1.3    ATTENTION ON THE AUGMENTED FEATURES:

The resulting features, $\mathbf{F}''$, were then fed into a transformer with EdgeConv as the query operation. Recall that EdgeConv (Wang et al., 2019b) computes graph features for each point using the equation:

$$\mathbf{e}_i \in \mathbb{R}^{d_e} = \max_{j \in \mathcal{N}(\mathbf{p}_i)}(\phi_{edge}(\mathbf{p}_i, \mathbf{p_j} - \mathbf{p}_i)), \tag{10}$$

where $\phi_{edge}$ is an MLP with output dimension $d_e$. The $\mathbf{F}''$ were then transformed using attention Vaswani et al. (2017):

$$\mathbf{Q} \in \mathbb{R}^{N \times d_k} = \text{EdgeConv}\,(\mathbf{F}'')\,W_q$$
$$\mathbf{K} \in \mathbb{R}^{(N \times k) \times d_k} = \text{Flatten}\,(\mathbf{F}'')\,W_k \tag{11}$$
$$\mathbf{V} \in \mathbb{R}^{(N \times k) \times d_v} = \text{Flatten}\,(\mathbf{F}'')\,W_v,$$

where $\mathbf{W}_q \in \mathbb{R}^{d_e \times d_k}$, $\mathbf{W}_k \in \mathbb{R}^{(2d_{in} + d_h) \times d_k}$ and $\mathbf{W}_v \in \mathbf{R}^{(2d_{in} + d_h) \times d_v}$ are learnable weight matrices. Our final point-level output features from the transformer block was then given by:

$$\mathbf{z}_i \in \mathbb{R}^{N \times d_v} = \mathbf{q}_i(\text{softmax}(\mathbf{k}_i)^\top \mathbf{v}_i). \tag{12}$$

For all experiments, we used two transformer layers such that the final feature vector for each point was of size 256.

### A.2    CURVENET ADAPTATION

CurveNet uses sampling and grouping. Our only adaptation to CurveNet was use the same number of input points as input into the farthest point sampling algorithm. We kept everything else the same as the original paper. We replaced the original adaptive max, adaptive mean pooling, and the classification head with MIL pooling. The final feature vector for each point was of size 1024.

### A.3 POINTNEXT ADAPTATION

PointNeXt uses sampling and grouping. To adapt PointNeXt to POINTMIL, we did not modifying the architecture itself. Instead, we concatenated the point-level features from the first layer of the encoder with global features from the final layer of the encoder. This resulted in a final feature vector for each point of size 544.

### A.4 MIL POOLING

#### A.4.1 CLASSIFICATION HEAD

We tested several different classification heads for each dataset. The final classification heads for each dataset are summarised in Table 4.

Table 4: Classification head architecture

| Type | Layer | Input | Output |
|---|---|---|---|
| IntrA/RBC | Linear | $b \times 1 \times N \times d$ (feature dimension) | $b \times 1 \times N \times c$ |
| MN40 | Linear + ReLU | $b \times 1 \times N \times d$ | $b \times 1 \times N \times d//2$ |
| | Linear + ReLU | $b \times 1 \times N \times d//2$ | $b \times 1 \times N \times d//4$ |
| | Linear | $b \times 1 \times N \times d//4$ | $b \times 1 \times N \times c$ (Point Pred) |

#### A.4.2 ATTENTION HEAD

Table 5: Attention head architecture

| Process | Layer | Input | Output |
|---|---|---|---|
| Attention | Linear + tanh | $b \times 1 \times N \times d$ | $b \times 1 \times N \times 8$ |
| | Linear + sigmoid | $b \times 1 \times N \times 8$ | $b \times 1 \times N \times 1$ (Attn. Scores) |

We used the same attention head for all attention-based pooling. This is summarised in Table 5.

## B INTERPRETABILITY METRICS

AOPCR does not require instance labels, whereas NDCG@n does. AOPCR works by removing the most important instances in sequence and observing the impact on prediction accuracy. The faster the prediction declines, the better the ordering, as the most influential instances are removed earlier. When point clouds are annotated, NDCG@n evaluates how closely the model's interpretability ranking matches the true order. It rewards rankings that prioritise relevant instances, with higher scores indicating better alignment and interpretability.

## C DATASETS

### C.1 INTRA

IntrA is an open source dataset of 3D intracranial aneurysm (Yang et al., 2020). The task is to classify blood vessels as healthy and aneurysm. There is a total of 1909 blood vessel segments, including 1694 healthy vessel segments and 215 aneurysm segments for diagnosis. 116 of the aneurysm segments are expertly annotated. We use IntrA to evaluate interpretability, classification, and segmentation.

### C.2 RED BLOOD CELL

We used another dataset of 3D red blood cells (RBC; Simionato et al. (2021)) for classification. This dataset includes 825 3D red blood cells imaged using confocal microscopy grouped into 9 expertly

annotated shape classes. Blood samples were collected from healthy donors and patients using finger prick blood sampling. For inducing RBC shape transitions, blood from 5 healthy donors was treated with NaCl solutions of varying concentrations to create different RBC shapes. Specific shape classes were expertly annotated according to particular motifs. Thus, similar to IntrA, RBC was suitable for evaluating interpretability by the ability to identify these motifs. Segmentation masks are publicly available. We converted the segmentation to mesh objects using marching cubes with Laplacian smoothing, and then sampled points from the vertices of these mesh objects.

### C.3 MODELNET40

ModelNet40 (Wu et al., 2015) is the *de-facto* benchmark for point cloud classification containing 9,843 training and 2,468 testing meshed CAD models belonging to 40 different object classes.

### C.4 SHAPENETPART

ShapeNetPart (Yi et al., 2016) consists of 16,881 shapes with 16 classes belonging to 50 parts labels. We use ShapeNetPart for segmentation.

### C.5 TRAINIG SPLITS

For IntrA and RBC, we used a five-fold cross-validation and reported the average test metrics across folds. For ModelNet40 and ShapeNetPart, we used the provided train and test splits and reported the test results.

## D ADDITIONAL RESULTS

This section contains additional results of individual pooling methods.

### D.1 INTERPRETABILITY

Tables 6, 7, and 8 show the IntrA interpretability results for each of the pooling methods using the Transformer, PointNet, and DGCNN backbones, respectively. The mean and standard deviations on the test sets across the five folds are shown.

Table 6: Additional POINTMIL interpretability results on IntrA using the transformer backbone. We also show the effect of the best contextual attention for each attention-based method.

| Model | NDCG@n | AOPCR |
|---|---|---|
| Additive | $0.613_{0.033}$ | $18.108_{4.374}$ |
| Additive + context 12 | $0.608_{0.035}$ | $18.162_{3.013}$ |
| Attention | $0.426_{0.030}$ | $10.336_{1.065}$ |
| Attention + context 12 | $0.539_{0.019}$ | $14.541_{1.821}$ |
| Conjunctive | $0.592_{0.018}$ | $12.526_{2.960}$ |
| Conjunctive + context 12 | $0.610_{0.024}$ | $16.305_{5.859}$ |
| Instance | $0.587_{0.022}$ | $16.166_{3.794}$ |

Table 7: Additional interpretability results on IntrA using POINTMIL with the PointNet backbone

| Model | NDCG@n | AOPCR |
|---|---|---|
| Additive | $0.254_{0.064}$ | $0.792_{0.298}$ |
| Attention | $0.222_{0.027}$ | $0.005_{0.035}$ |
| Instance | $0.225_{0.072}$ | $0.973_{0.212}$ |
| Conjunctive | $0.208_{0.067}$ | $0.741_{0.140}$ |

Table 8: Additional interpretability results on IntrA using POINTMIL with the DGCNN backbone

| Model | NDCG@n | AOPCR |
|---|---|---|
| Additive | $0.482_{0.009}$ | $4.486_{0.550}$ |
| Attention | $0.223_{0.002}$ | $-0.031_{0.070}$ |
| Conjunctive | $0.467_{0.008}$ | $4.828_{0.617}$ |
| Instance | $0.462_{0.022}$ | $5.212_{0.547}$ |

# E VISUAL INTERPRETATION EXAMPLES

Figure 10 shows additional interpretability visualisations on ModelNet40.

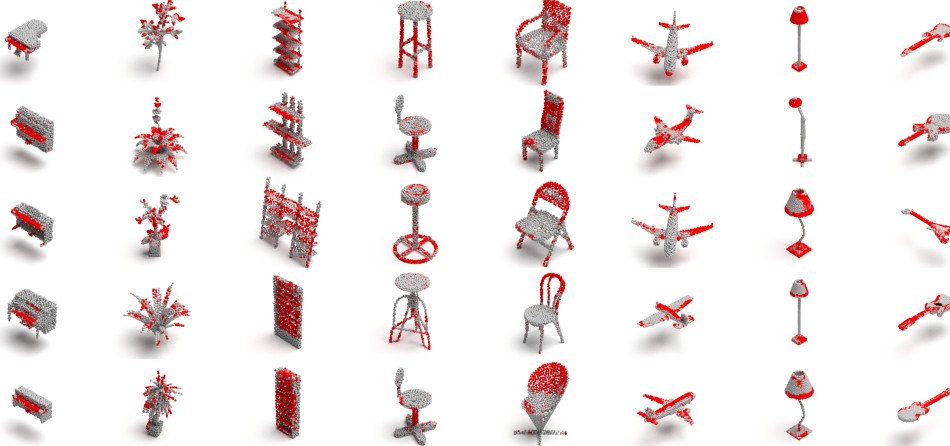

Figure 10: Examples of POINTMIL interpretations for correctly classified shapes from ModelNet40.

## F ROBUSTNESS TO NOISE

Similar to the methods described by
Xiang et al. (2021) and Yan et al.
(2020), we assessed the robustness of
POINTMIL to noisy inputs. Specif-
ically, we measured the F1 score of
models trained on clean (raw) inputs
when subjected to noisy inputs dur-
ing inference. This approach allowed
us to evaluate the model's ability to
maintain performance in the presence
of input perturbations. The F1 score

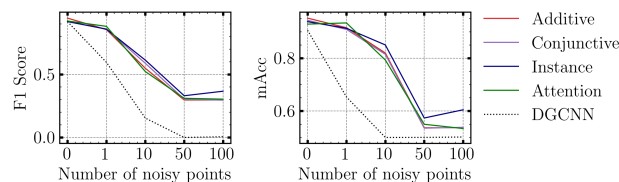

Figure 11: Robustness evaluation of models to noisy inputs.

(left) and the mACC (right) is plotted against the number of noisy points introduced during inference
for different POINTMIL methods with the DGCNN backbone and the original DGCNN model in
Figure 11. POINTMIL methods demonstrate higher robustness to noise compared to baseline mod-
els, with `Additive` and `Conjunctive` maintaining consistently higher F1 and mACC scores
than the original DGCNN without MIL.

## G SEGMENTATION

Figure 12 presents segmentation results for POINTMIL with the Transformer backbone in the IntrA
dataset. Clearly, POINTMIL is able to accurately Aneurysm regions with a 3D shape of a diseased
blood vessel.

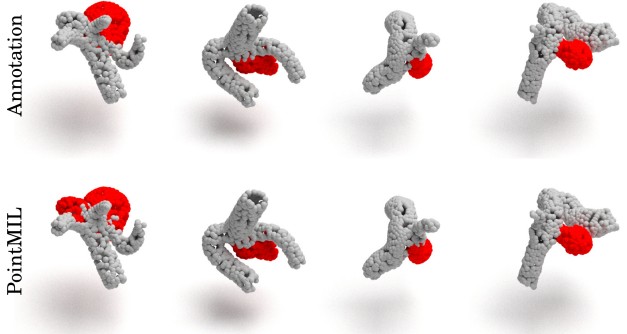

Figure 12: Segmentation examples for POINTMIL with the Transformer backbone on the IntrA
dataset.

## H RENDERING

All renderings of point clouds were made with Mitsuba2.