# OpenReview forum: "Interpretable point cloud classification using multiple instance learning"
_ICLR.cc/2025/Conference — ICLR 2025 Conference Withdrawn Submission_

### Official Review · Reviewer_CZ3U · 2024-10-23

**Soundness:** 2
**Presentation:** 3
**Contribution:** 2
**Rating:** 5
**Confidence:** 3

**Summary:**

This paper introduces multiple instance learning to provide local interpretability for point cloud classification models. Experiments are conducted on several popular point cloud classification models and classification benchmarks, demonstrating the effectiveness of the proposed method.

**Strengths:**

This paper is the first to introduce multiple instance learning for interpretable learning in the 3D domain. It can be integrated as a simple module into existing point cloud classification models, effectively enhancing classification performance while also improving interpretability.

**Weaknesses:**

1.The point cloud classification models applied in PointMIL appear to be relatively outdated, as more advanced models such as PointNeXt and PointMLP, as well as more classic models like PointNet++, have not been included in the experiments.

2.The practicality of PointMIL in real-world applications remains questionable, as it lacks experiments on real datasets such as ScanObjectNN, or robustness tests against noise, rotation, and other transformations.

3.There are some minor errors in the details: $\textbf{(a)}$. it seems that $\hat{y}$ in Equation 8 is not defined. $\textbf{(b)}$. there is an error in Table 4 of the appendix

**Questions:**

1.Is the performance comparison on classification tasks fair? If I understand correctly, PointMIL eliminates downsampling to achieve point-level predictions, whereas the other methods compared, such as PointNet++, still employ downsampling. The authors should consider the impact of downsampling on performance.

2.I recommend that the authors include experimental results for PointMIL on ScanObjectNN and test the interpretability on more advanced point cloud models such as PointNeXt and PointMLP.

3.Is PointMIL also applicable to scene-level tasks, such as scene segmentation that requires generating point-level predictions? This could effectively demonstrate the applicability of PointMIL.


4. I hope the authors can provide a detailed explanation of the results, for example, why $\textbf{Additive}$ and $\textbf{Conjunctive}$ achieve better results compared to other MIL pooling methods in Figure 2. Such an exploration would be beneficial for understanding how to choose MIL pooling methods under different circumstances.

---

> ### Author Response · Authors · 2024-11-25
> **Response to Reviewer CZ3U (Part 1)**
>
> We thank the reviewer for their positive comments and constructive criticism. In particular, we thank the reviewer for pointing out the novelty of our work. Below we address the weaknesses and questions.
>
> ### **Newer backbones:**
> >1. The point cloud classification models applied in PointMIL appear to be relatively outdated, as more advanced models such as PointNeXt and PointMLP, as well as more classic models like PointNet++, have not been included in the experiments.
>
> We appreciate this comment and agree that it is highly necessary for us to demonstrate PointMIL on more backbones. Therefore, to directly address this, we have extended our experiments to include an additional modern backbone, CurveNet [1], as discussed in Section 3.1. CurveNet is a widely adopted architecture that builds on recent advancements in capturing point cloud geometry through curve learning. We specifically chose CurveNet as the adaptations that were done to this method could be applied to any other method using farthest point sampling, including but not limited to PointMLP [2], and PointNeXt [3]. This resulted in four new models. Importantly, we adapted CurveNet to retain per-point features by removing point downsampling steps, ensuring compatibility with MIL pooling and enabling point-specific interpretability. This adaptation demonstrates the flexibility of PointMIL in integrating with modern architectures. We have included the interpretability results of CurveNet in Table 1, showing that PointMIL outperforms CLAIM and PSM in both APOCR and NDCG@n for the CurveNet backbone. Please see the table below:
>
> | Method       | AOPCR   | NDCG@n  |
> |--------------|---------|---------|
> | PSM          | 1.371   | 0.218   |
> | CLAIM        | 1.363   | 0.252   |
> | **Additive**     | 0.615   | **0.266**  |
> | **Attention**     | 1.52    | 0.260 |
> | **Conjunctive**    | **2.66**    | 0.207   |
> | **Instance**    | 1.709 | 0.236   |
>
>
> We also show that PointMIL improves accuracy of CurveNet. To fairly demonstrate this, we present the original CurveNet architecture, the adapted architecture without MIL pooling, using the same classification head in the original paper, and then the adapted CurveNet with MIL pooling. The PointMIL version outperforms the original and adapted CurveNet on both real-world biological datasets by up to 3% mACC. Regarding ModelNet40, the PointMIL version of CurveNet outperforms the adapted version by 0.1%. However, the original CurveNet outperforms PointMIL by 0.3%. The results on the three datasets are shown below. The adapted CurveNet architecture is shown with a $^{\dagger}$ .
>
> | Method                                | IntrA mACC (↑) | IntrA F1 (↑) | RBC mACC (↑) | RBC F1 (↑) | ModelNet40 mACC (↑) | ModelNet40 oACC (↑) |
> |---------------------------------------|----------------|--------------|--------------|------------|---------------------|---------------------|
> | CurveNet                              | 88.3           | 89.8         | 88.3         | 87.8       | -                   | 93.8                |
> | CurveNet$^{\dagger}$                  | 87.8           | 87.8         | 85.8         | 85.7       | 90.6                | 93.4                |
> | PointMIL (CurveNet$^{\dagger}$)       | 91.3 (+3.5)    | 89.9 (+2.1)  | 91.2 (+5.4)  | 90.5 (+4.8)| 91.0 (+0.4)         | 93.5 (+0.1)         |
>
> We chose CurveNet as the adaptations that were done to this method could be applied to any other method using farthest point sampling, including but not limited to PointNeXt, PointMLP, and PontNet++. We also with to clarify that applying PointMIL to an extra backbone is adding another four models as there are four versions of PointMIL. Therefore, for this rebuttal prior we focussed on applying PointMIL to an architecture representative of the most recent architectures.
>
> [1] Tiange Xiang, et al. Walk in the cloud: Learning curves for point clouds shape analysis. In Proceedings of the IEEE/CVF International Conference on Computer Vision (ICCV), pp. 915–924, October 2021.
>
> [2]Xu Ma, Can Qin, Haoxuan You, Haoxi Ran, and Yun Fu. Rethinking network design and local geometry in point cloud: A simple residual MLP framework. In International Conference on Learning Representations, 2022.
>
> [3] Guocheng Qian, et al. Pointnext: Revisiting pointnet++ with improved training and scaling strategies. In S. Koyejo, S. Mohamed, A. Agarwal, D. Belgrave, K. Cho, and A. Oh (eds.), Advances in Neural Information Processing Systems, volume 35, pp. 23192–23204. Curran Associates, Inc., 2022.

---

> ### Author Response · Authors · 2024-11-25
> **Response to Reviewer CZ3U (Part 2)**
>
> >2.The practicality of PointMIL in real-world applications remains questionable, as it lacks experiments on real datasets such as ScanObjectNN, or robustness tests against noise, rotation, and other transformations.
>
> ### **Real world application:**
> Thank you for your comment. We would like to clarify that our work extensively evaluates PointMIL on two real-world datasets: IntrA and RBC, both of which represent practical applications in biomedical imaging. Specifically:
> * On the IntrA [1] dataset, which involves intricate, real-world biomedical point clouds of vascular structures, PointMIL achieved state-of-the-art performance, surpassing all prior methods in both classification accuracy and interpretability metrics.
>
> * On the RBC [2] dataset, consisting of red blood cell images, PointMIL similarly demonstrated state-of-the-art results, validating its effectiveness in real-world scenarios beyond synthetic benchmarks.
>
> Additionally, we evaluated PointMIL on widely-used benchmark datasets—ModelNet40 and ShapeNetPart—to ensure its comparability with existing point cloud methods. On these benchmarks, PointMIL achieved competitive results, further demonstrating its robustness and generality across diverse datasets. While we were unable to present results on another dataset as this involves training 16 PointMIL models (four MIL pooling methods on four backbones) as well as reproducing the other state-of-the-art, we believe that we have already demonstrated PointMIL’s application to real world datasets. This was a specific reason for choosing IntrA and RBC. Furthermore, these datasets have distinctive salient 3D shape motifs that we could assess model interpretability on.
>
> ### **Robustness to noise**:
> To address your point about robustness to noise, we conducted experiments to evaluate how each model performs under noisy conditions during inference. Specifically, the models were trained on raw, unperturbed point clouds but tested on perturbed point clouds, where noise was added to a varying subset of points. We follow a similar approach to [3, 4]. This is presented in the Appendix Figure 8 of the updated manuscript. We saw that PointMIL (with Transformer backbone) outperforms PointMLP, PointNet, and DGCNN, demonstrating robustness to noise on IntrA.
>
> [1] Xi Yang, et al. Intra: 3d intracranial aneurysm dataset for deep learning. In Proceedings of the IEEE/CVF Conference on Computer Vision and Pattern Recognition (CVPR), June 2020.
>
> [2] Greta Simionato, et al. Red blood cell phenotyping from 3d confocal images using artificial neural networks. PLOS Computational
> Biology, 17(5):1–17, 05 2021
>
> [3] Mingye Xu, et al. Geometry sharing network for 3d point cloud classification and segmentation. In AAAI, pages 12500–12507, 2020
> [4] Tiange Xiang, et al. Walk in the cloud: Learning curves for point clouds shape analysis. In Proceedings of the IEEE/CVF International Conference on Computer Vision (ICCV), pp. 915–924, October 2021.
>
> ### **Minor errors:**
> >There are some minor errors in the details: (a). it seems that $\hat{\mathbf{y}}$ in Equation 8 is not defined. (b). there is an error in Table 4 of the appendix.
>
> We appreciate the close attention to our paper. $\hat{\mathbf{y}}$ should have been $\mathbf{z_i}$ in this equation. Table 4 originally showed Additive + context 12 twice. The second time, this should have said Attention + context 12. We have now updated both of these.

---

> > ### Comment · Reviewer_CZ3U · 2024-11-25
> > **Question About Real world application**
> >
> > Thank you for your response. I would like to know if there are any other synthetic medical imaging datasets besides the two real medical imaging datasets, IntrA and RBC, and what the differences are between them. The notable difference between ScanObjectNN and ModelNet40 is that the point clouds in ScanObjectNN are affected by factors such as occlusion, noise, and background. Addressing these questions would help in understanding the feasibility of PointMIL in real-world applications.

---

> > ### Comment · Reviewer_CZ3U · 2024-11-27
> > **Question About  **Robustness to noise****
> >
> > Thank you for your response and revisions. I have concerns regarding experiment  **Robustness to noise**, as previous work[1] has demonstrated that Transformers exhibit superior robustness when facing factors such as noise and rotation. Therefore, the contribution of PointMIL in this experiment is questionable. A better approach would be to apply PointMIL to architectures like DGCNN and compare its robustness with the original DGCNN.
> >
> > [1] Sun, Jiachen, et al. "Benchmarking robustness of 3d point cloud recognition against common corruptions." arXiv preprint arXiv:2201.12296 (2022).

---

> > > ### Author Response · Authors · 2024-11-30
> > > **Request review of recent updates**
> > >
> > > Dear Reviewer CZ3U,
> > >
> > > As the discussion period deadline is fast approaching, we kindly wish to follow up regarding your feedback on our recent updates. To summarise, we have:
> > > * completed experiments on both backbones that you suggested, PointMLP and PointNeXt as well as an additional third, CurveNet. All of which PointMIL improves. This showcases its adaptability to modern architectures.
> > > * conducted the robustness to noise experiment showing that PointMIL improves robustness to noise on DGCNN (Figure 11). We also showed how interpretations are robust to noise (Figure 9).
> > >
> > > We truly appreciate your time and effort in reviewing our submission and look forward to hearing from you.
> > >
> > > Best regards, Authors

---

> > > > ### Comment · Reviewer_CZ3U · 2024-12-02
> > > >
> > > > Thank you for your additions; they have certainly improved the quality of the paper to some extent. However, we believe that the method requires certain modifications to the model architecture and retraining to achieve a level of interpretability, and its practical value remains questionable. Additionally, the innovation is limited, as it primarily adopts existing multi-instance learning techniques. Therefore, we have decided to maintain the  score.

---

> ### Author Response · Authors · 2024-11-25
> **Response to Reviewer CZ3U (Part 3)**
>
> ### **Questions:**
>
> >1. Is the performance comparison on classification tasks fair? If I understand correctly, PointMIL eliminates downsampling to achieve point-level predictions, whereas the other methods compared, such as PointNet++, still employ downsampling. The authors should consider the impact of downsampling on performance.
>
> Thank you for raising this point. We would like to clarify that while PointMIL does not employ downsampling, this does not inherently provide an advantage in classification tasks. In fact, methods that incorporate sampling and grouping mechanisms, such as PointNet++, PointMLP, PointNeXt and CurveNet, often outperform those that do not. For example, CurveNet leverages sampling and grouping to capture local geometric structures more effectively, which contributes to its high performance on benchmark tasks. In our updated version of the manuscript we have included the CurveNet backbone. However, we had to adapt it to not employ sampling and grouping. Table 2 shows how across datasets, the version of CurveNet that does not include sampling and grouping (does not downsample points), actually performs worse than the original CurveNet that does downsample points. We further show that then adding MIL pooling onto the adapted CurveNet, we increase performance back beyond that of the original architecture.
>
> It is important to note that downsampling is a component of the architecture design, not of the data that is fed into the model. PointNet++ and similar architectures use downsampling and grouping to improve their feature extraction and overall performance. In contrast, PointMIL does not use downsampling because its primary goal is to provide point-level predictions and inherent interpretability, which requires maintaining per-point granularity throughout the pipeline.
>
> We emphasise that comparing PointMIL to downsampling-based methods is fair because it highlights the trade-off between pure classification performance and interpretability. While PointMIL prioritises maintaining per-point granularity to provide inherent interpretability, methods with downsampling aim to optimise classification performance by leveraging sampling and grouping techniques. This comparison showcases the distinct strengths and applications of both approaches, and we hope this clarification addresses your concern. Thank you for the opportunity to elaborate on this aspect of our work.
>
> >2. I recommend that the authors include experimental results for PointMIL on ScanObjectNN and test the interpretability on more advanced point cloud models such as PointNeXt and PointMLP.
>
> We completely agree with the reviewer regarding more advanced models. Please see our response in Part 1 above showing our analysis on CurveNet [1]. Regarding real world datasets, please see our response in Part 2 of our response.
>
>
> >3.Is PointMIL also applicable to scene-level tasks, such as scene segmentation that requires generating point-level predictions? This could effectively demonstrate the applicability of PointMIL.
>
> Thank you for this thoughtful suggestion. PointMIL is inherently designed to provide point-level predictions, making it conceptually applicable to scene-level tasks such as scene segmentation. However, our primary focus in this work has been on point cloud classification with inherent interpretability, which aligns with the challenges presented by the datasets evaluated (e.g., IntrA and RBC).
>
> Scene segmentation presents additional challenges, such as handling large-scale point clouds and complex spatial relationships, which could require further adaptations of the PointMIL framework, particularly in terms of scalability and computational efficiency. While we have not evaluated PointMIL on scene segmentation tasks in this study, its ability to generate point-level predictions and integrate interpretability mechanisms suggests strong potential for such applications.
>
> Future work could focus on transforming scene level segmentation tasks into a multiple instance learning classification problem where one could classify whether or not there is a chair in the scene and then use interpretations to determine whether the salient regions are indeed of a chair.
>
> We have noted this as an exciting avenue for future research and appreciate your suggestion to explore scene-level tasks to further demonstrate the generalisability of PointMIL. Thank you for your constructive feedback.
>
>
> [1] Tiange Xiang, et al. Walk in the cloud: Learning curves for point clouds shape analysis. In Proceedings of the IEEE/CVF International Conference on Computer Vision (ICCV), pp. 915–924, October 2021.

---

> ### Author Response · Authors · 2024-11-25
> **Response to Reviewer CZ3U (Part 4)**
>
> ### **Questions (ctd)**:
>
> >I hope the authors can provide a detailed explanation of the results, for example, why Additive and Conjunctive achieve better results compared to other MIL pooling methods in Figure 2. Such an exploration would be beneficial for understanding how to choose MIL pooling methods under different circumstances.
>
> Thank you for your insightful comment. We have now expanded the discussion in the manuscript to provide a detailed explanation of why additive and conjunctive pooling achieve better results compared to other MIL pooling methods, as shown in Figure 2.
>
> * **Additive pooling:** combines the importance scores (attention weights) with per-point feature representations and then classifies on these weighted features. This allows the model to retain more information about the individual points’ contributions, providing a balance between interpretability and classification accuracy. By scaling feature importance across all points, additive pooling ensures that the model can focus on both dominant and subtle patterns in the data, which is particularly advantageous in complex point clouds.
>
> * **Conjunctive pooling** goes a step further by computing attention weights and point-level predictions independently and then combining them. This mechanism allows for class-specific explanations by explicitly linking importance scores to the final classification decision only after the point-level classifications have been made. This allows the point-level classification to be made regardless of the importance. As a result, conjunctive pooling achieves superior interpretability and accuracy because it can focus on the most relevant points for a given class, while also preserving global context.
>
> **Why these perform better:**
>
> Both additive and conjunctive pooling outperform simpler pooling methods (e.g., instance and attention pooling) because they integrate additional layers of context:
> * Instance pooling treats points equally, which limits the model’s ability to capture the most informative points in a point cloud.
> * Attention pooling provides importance scores but lacks the class-specific focus that conjunctive, additive, and instance pooling achieves.
>
> The additional expressiveness of additive and conjunctive pooling, combined with the neighbourhood smoothing provided by contextual attention, enhances both interpretability and classification performance.
>
> We have updated our manuscript to say;
>
> "This superior performance of \texttt{Additive} and \texttt{Conjunctive} pooling can be attributed to their ability to better aggregate point-level importance scores. Additive pooling scales point features with their importance weights, preserving detailed information while focusing on relevant points. Conjunctive pooling further enhances this by independently computing attention weights and class-specific contributions, explicitly aligning the model's focus with the predicted class. In contrast, Instance pooling lacks this importance weighting, and Attention pooling does not offer class-specific explanations and rather provides a general measure of importance across classes, which limits their interpretability.”

---

> ### Author Response · Authors · 2024-11-25
> **Response to Question About Real world application**
>
> Thank you for quick follow-up.
>
> Currently, there are no widely-used synthetic medical imaging datasets for 3D point clouds comparable to ModelNet40 or ScanObjectNN in terms of scale and diversity, that we know of. Most medical imaging datasets, including IntrA and RBC, are derived from real-world data. The differences between IntrA and RBC are as follows:
>
> * **IntrA [1]:** This dataset consists of point clouds of vascular structures extracted from biomedical images. It presents challenges such as intricate 3D shapes, local deformations, and fine-grained structural details, requiring models to focus on local regions for accurate classification and interpretation. Importantly, there exist specific local shape motifs that (the region of Aneurysm) that are directly related to the classification. Therefore, interpretability is trivial to evaluate. The data was acquired by reconstructing scanned 2D MRA images of patients, with healthy and diseased segments automatically generated.
>
> * **RBC [2]:** Blood samples were collected from healthy donors and patients using finger prick blood sampling. For inducing red blood cell (RBC) shape transitions, blood from 5 healthy donors was treated with NaCl solutions of varying concentrations to create different RBC shapes. cells imaged using confocal microscopy. Specific shape classes were expertly annotated according to particular motifs. Thus, similar to IntrA, RBC was suitable for evaluating interpretability by the ability to identify these motifs. Segmentation masks are publicly available. We converted the segmentation to mesh objects using marching cubes with Laplacian smoothing, and then sampled points from the vertices of these mesh objects.
>
> We have updated our appendix to discuss these datasets in more detail.
>
>
> As far as we know, one sythetic dataset of migrating T Cells [3] exists, however, this has not been affeted by noise, occlusion, and background factors.
>
> However, we have evaluated the effect of noisy samples on the IntrA dataset and compared PointMIL to PointMLP, DGCNN, and PointNet (Figure 8 in Appendix). This showed that PointMIL is considerably more robust to noise than these other methods. We believe that this is a particular strength of attention-based MIL pooling, where the attention mechanism allows the model to focus on informative points and ignore noisy or redundant information.
>
> For future work, we would definitely like to assess PointMIL on ScanObjectNN. However, this was out of the scope of this paper as this would involve training 16 PointMIL models on another dataset. We hope that our analysis on noisy input and our demonstration of PointMIL's robustness to noise was sufficient in addressing this concern.
>
> [1] Xi Yang, et al. Intra: 3d intracranial aneurysm dataset for deep learning. In Proceedings of the IEEE/CVF Conference on Computer Vision and Pattern Recognition (CVPR), June 2020.
>
> [2] Greta Simionato, et al. Red blood cell phenotyping from 3d confocal images using artificial neural networks. PLOS Computational Biology, 17(5):1–17, 05 2021. doi: 10.1371/journal.pcbi.1008934
>
> [3] Medyukhina, A., et al. Dynamic spherical harmonics approach for shape classification of migrating cells. Sci Rep 10, 6072 (2020). https://doi.org/10.1038/s41598-020-62997-7

---

> ### Author Response · Authors · 2024-11-26
> **Additional backbone (PointNeXt, as suggested) in addition to CurveNet**
>
> >1.The point cloud classification models applied in PointMIL appear to be relatively outdated, as more advanced models such as PointNeXt and PointMLP, as well as more classic models like PointNet++, have not been included in the experiments.
>
>
> **In addition to our entension to CurveNet, we have now included PointNeXt [1], as per the reviewers suggestion** (an extra 2 during the rebuttal period bringing the total backbones to 5 and the total models to 20).
>
> PointNeXt uses sampling and grouping, however, this architecture required minimal adaptation for integration with PointMIL, due to the first layer outputting point-level features. Specifically, we concatenated point-level features from the first encoder layer with the global features from the last encoder layer to enable MIL pooling without modifying the architecture itself. This approach demonstrates the adaptability of PointMIL to newer backbones while preserving architectural integrity.
>
> Our interpretability results, shown below, highlight that **PointMIL outperforms baseline methods such as CLAIM and PSM on both AOPCR and NDCG@n metrics for PointNeXt**, further validating its effectiveness. This is also shown in Table 1 of the revised manuscript.
>
> | Method         | AOPCR   | NDCG@n  |
> |----------------|---------|---------|
> | PSM            | 0.092   | 0.272   |
> | CLAIM          | 0.226   | 0.294   |
> | **Additive**       | 1.259   | 0.300   |
> | **Attention**       | 0.044   | 0.235   |
> | **Conjunctive**     | 1.531   | **0.310**   |
> | **Instance**       | **2.160**   | 0.285   |
>
> We also evaluated PointMIL's impact on classification performance with PointNeXt. As shown below, **PointMIL improves PointNeXt's classification accuracy on real-world datasets like IntrA and RBC, achieving up to a 2.8% increase in mACC on IntrA.** On ModelNet40, PointMIL maintains competitive performance, with only a 0.3% decrease in mACC and 0.1% increase in oACC compared to the original PointNeXt. While the improvements on ModelNet40 are minor, it is important to point out that this is the first inherently interpretable framework for point cloud analysis. There often exists a trade-off between interpretation and classification, however **PointMIL offered interpretability without harming and often improving classification performance.**
>
> | Method                             | IntrA mACC (↑) | IntrA F1 (↑) | RBC mACC (↑) | RBC F1 (↑) | ModelNet40 mACC (↑) | ModelNet40 oACC (↑) |
> |------------------------------------|----------------|--------------|--------------|------------|---------------------|---------------------|
> | PointNeXt                          | 91.8           | 94.7         | 86.1         | 87.1       | 90.8                | 93.2                |
> | **PointMIL (PointNeXt)**           | **94.6** (+2.8)| **96.2** (+1.5)| **87.6** (+1.5)| **88.2** (+0.4)| **90.5** (-0.3)     | **93.3** (+0.1)     |
>
> These results demonstrate the robustness and generalisability of PointMIL across diverse datasets, from biological data (IntrA and RBC) to everyday objects (ModelNet40). The inclusion of PointNeXt and CurveNet underscores the adaptability of PointMIL to contemporary architectures, highlighting its utility as the first inherently interpretable framework for point cloud classification. This work lays the foundation for future research in 3D point cloud interpretability, bridging the gap between performance and interpretability in this domain.
>
> [1] Guocheng Qian, et al. Pointnext: Revisiting pointnet++ with improved training and scaling strategies. In S. Koyejo, S. Mohamed, A. Agarwal, D. Belgrave, K. Cho, and A. Oh (eds.), Advances in Neural Information Processing Systems, volume 35, pp. 23192–23204. Curran Associates, Inc., 2022.

---

> ### Author Response · Authors · 2024-11-28
> **Author response to Question About **Robustness to noise** and Added third backbone**
>
> ### **Robustness to noise**
>
> Thank you for raising these concerns and offering a better approach. We have now performed those additional experiments by applying PointMIL to DGCNN and compared the robustness to noise with the original DGCNN on the IntrA dataset in terms of F1 score and mACC. This is now presented in Figure 11 in Appendix F of the updated manuscript. PointMIL outperformed the original DGCNN by substantial margins, demonstrating the robustness to noise.
>
> To further address this concern, we have provided visualisations of interpretations of PointMIL using Transformer backbone on an Airplane from ModelNet40, shown in Figure 9 in the main of the updated manuscript. Figure 9 shows how, even when noisy
> points are added to objects, PointMIL is still able to focus on salient 3D shape motifs, ignoring the noise.
>
> We hope that these two analyses have addressed your concern.
>
> ### **PointMLP backbone**
> We have added a third additional backbone to our analysis. As per the reviewers suggestion, we applied PointMIL to PointMLP. Specifically, we applied PointMIL to PointMLPElite by adapting the architecture not downsample the points. We showed the affect of this adaptation on classification results so that any difference in performance can then be attributed to the MIL pooling instead of this adaptation. We ran these experiments on the ModelNet40 dataset. The results are shown in Table 2 of the updated manuscript and in the table below:
> | Method                                  | mACC(↑) | oACC(↑) |
> |-----------------------------------------|---------|---------|
> | PointMLP            | **91.3** | **94.1** |
> | PointMLPElite                           | 90.9    | 93.6    |
> | PointMLPElite$^{\dagger}$               | 90.1    | 92.6    |
> | PointMIL (PointMLPElite$^{\dagger}$) | 90.5 (+0.4) | 93.5 (+0.9) |
>
> The results show how removing downsampling originally decays the accuracy. However, applying PointMIL to the adapted architecture improves accuracy back to near the original version. Importantly **this model is now inherently interpretable**.
>
> We further show how the PointMIL version of PointMLP has similar regions of interpretations to DGCNN, CurveNet, and Transformer on an example Bed and Plant from the ModelNet40 dataset. This is shown in Figure 6 of the updated manuscript.
>
> **We sincerely hope that this work has addressed all your initial concerns and we look forward to hearing your response.**

---

> ### Author Response · Authors · 2024-12-02
> **Experiments on ScanObjectNN to demonstrate practical value**
>
> ### **ScanObjectNN**
>
> Thank you for your response. To address your concerns on the practical value of PointMIL, we have conducted experiments on the hardest version of ScanObjectNN (PB_T50_RS) for DGCNN, PointMLPElite, and Transformer backbones. Please see the table below:
>
> | Model                         | oACC         | mAcc          |
> |-------------------------------|--------------|---------------|
> | DGCNN (original)              | 78.1         | 73.6          |
> | DGCNN (PointNeXt training)    | 86.0 ± 0.5   |               |
> | PointMLPElite (original)      | 83.8 ± 0.6   | 81.8 ± 0.8    |
> | **PointMIL (DGCNN)**              | 85.6         | 83.0          |
> | **PointMIL (PointMLPElite)**      | 83.0         | 81.4          |
> | **PointMIL (Transformer)**        | 83.2         | 80.7          |
>
> We conducted these experiments to directly address the question of PointMIL's feasibility in real-world applications, specifically under the challenging conditions of ScanObjectNN, which includes occlusions, noise, and background clutter. As shown in the table:
> - **PointMIL (DGCNN)** (trained with PointNeXt strategies) achieved an **overall accuracy (oACC) of 85.6%** and **mean accuracy (mAcc) of 83.0%**, which falls within the standard deviation range of the DGCNN (PointNeXt training) baseline.
> - **PointMIL (PointMLPElite)** also demonstrated strong performance, with PointMLP results comparable to or within the standard deviation range of their respective original baselines. This highlights PointMIL's adaptability to different backbone architectures.
> -  **PointMIL (Transformer)** (trained with PointNeXt strategies) demonstrated strong performance.
> - These results validate that PointMIL can maintain competitive classification accuracy on a real-world dataset with inherent complexities like those in ScanObjectNN further demonstrating practical value.
>
> Finally, the fact that **PointMIL transforms these models to become inherently interpretable is the major strength** for it's practical value in real-world scenarios where model interpretability is fundamental.
>
> ### **Model modifications and retraining**
> While we appreciate that PointMIL requires certain models to be adapted, this adaptation is minimal, and even so, we show improved performance after allying PointMIL to this slightly adapted model. Furthermore, as the model is **inherently** interpretable, meaning that interpretations are an inbuilt feature in the model, it will always require retraining. Future work could explore fixing the weights from the feature extractor and only training the MIL pooling (similar to other work using MIL for whole slide image classification [1, 2, 3], and other MIL pipelines).
>
> [1] Shao, et al. *TransMIL: Transformer based Correlated Multiple Instance Learning for Whole Slide Image Classification.* NeurIPS 2021
>
> [2] Fourkioti, et al. *CAMIL: Context-Aware Multiple Instance Learning for Cancer Detection and Subtyping in Whole Slide Images.* ICLR. 2024
>
> [3] Song, et al. *Analysis of 3D pathology samples using weakly supervised AI.* *Cell*. 2024

---

> ### Author Response · Authors · 2024-12-03
> **Request review of recent updates from reviewer CZ3U**
>
> Dear Reviewer CZ3U,
>
> We kindly request that you review our recent update on our experiments on ScanObjectNN. This directly addresses your concern on the practical value of PointMIL. We have now included all backbones you suggested as well as the dataset you suggested. We have additionally discussed why retraining is used in this study and how freezing feature extractor weights could be a future direction of study.
>
> We sincerely thank you for your time in reviewing our work and look forward to hearing from you.
>
> Best regards,
> Authors

---

### Official Review · Reviewer_yUX5 · 2024-10-29

**Soundness:** 3
**Presentation:** 2
**Contribution:** 2
**Rating:** 6
**Confidence:** 5

**Summary:**

This paper proposes a method for generating point-level importance in point cloud classification, aiming to interpret the contribution of each point to the classification outcome. A modified Transformer is introduced to extract point-level features, with MIL pooling applied to determine each point's importance. The paper presents a strong motivation and is well-organized.

**Strengths:**

The motivation is innovative, focusing on the interpretability of point cloud models and aiming to address the issue of poor interpretability in existing models.

**Weaknesses:**

1. The authors aim to propose a general interpretability method; however, they add a Transformer-based feature extractor to existing models as an additional modification to the backbone network. This modification can be seen as an alteration to the backbone, which may impact the original performance of the point cloud network. Consequently, conducting interpretability analysis on this modified version may not align with the original motivation.

2. The novelty is limited: the authors' contributions primarily include (1) the feature extractor and (2) MIL pooling. However, the core iead of the feature extractor is merely a straightforward combination of existing Transformer and DGCNN approaches, lacking significant originality. In the MIL pooling, existing ideas are directly applied without any improvements tailored to the inherent characteristics of point cloud data. These factors restrict the paper's novelty, as it largely reflects a simple combination and adaptation of existing methods.

3. The performance improvements are likely attributed mainly to the additional feature extraction network.

**Questions:**

Although the authors provide visualizations of points deemed important for classification in the figures, these points are derived from the final pooling output. Why are these highlighted red points considered more important—just because they have higher importance scores? The current conclusions remain insufficiently convincing. Are there alternative methods that could offer supplementary validation? One possible approach could involve removing the red points and observing a significant accuracy drop, while removing the gray points leads to no substantial accuracy decrease. However, this is an intuitive idea and may not hold in actual experiments.

---

> ### Author Response · Authors · 2024-11-25
> **Response to Reviewer yUX5 (Part 1)**
>
> We want to thank the reviewer for the constructive criticism. We address each weakness below:
>
> ### **Misunderstanding regarding modification to backbone**
>
> > The authors aim to propose a general interpretability method; however, they add a Transformer-based feature extractor to existing models as an additional modification to the backbone network. This modification can be seen as an alteration to the backbone, which may impact the original performance of the point cloud network. Consequently, conducting interpretability analysis on this modified version may not align with the original motivation.
>
> Thank you for raising this concern. We believe there is a misunderstanding regarding the role of the Transformer-based feature extractor in our work. To clarify, the backbones were **not** modified with a transformer-based feature extractor. We simply adapt the classification heads in the original architectures to replace them with MIL pooling heads **after** the feature extraction. The attention used in our MIL pooling is **not** a transformer feature extractor. We clarify this in detail below:
>
> 1. **Scope of Our Work:**
> Our primary aim is to propose a general interpretability method, not a backbone-specific enhancement. PointMIL is designed to work with a variety of existing backbone networks. We additionally propose a Transformer backbone. The inclusion of the Transformer-based feature extractor in our own Transformer backbone (not in all backbones) serves as one example to demonstrate the flexibility and adaptability of PointMIL to modern architectures. Importantly, this is not a requirement for PointMIL; the method can be applied to any backbone that outputs per-point features. In addition to our Transformer backbone, we originally demonstrated PointMIL on PointNet, and DGCNN backbones where no changes were made besides removing the classification heads of each original architecture and replacing it with MIL pooling. We did not include a transformer-based feature extractor to these models at all. We state in Section 3, that instead of averaging features across points before passing into a classifier, PointMIL utilises point-level features to obtain point-level classification and interpretation scores.
>
> 2. **Original Backbone Performance:**
> We ensure that the interpretability framework respects the original performance of the backbone by not altering its core functionality. For instance, in our experiments with DGCNN and PointNet, we maintained the original architectures and only integrated PointMIL's interpretability mechanisms without affecting the underlying feature extraction pipeline. For the Transformer-based backbone, we included it as an additional example to highlight PointMIL's compatibility with a modern architecture, rather than as a replacement for existing backbones. During this revision period, we have now included a fourth backbone, CurveNet. With CurveNet, the only adaptation that we made to the original architecture was the removal of farthest point sampling as we require point-level features from the feature extractor.
>
> This adaptation demonstrates the flexibility of PointMIL in integrating with modern architectures. We have included the interpretability results of CurveNet in Table 1, showing that PointMIL outperforms CLAIM and PSM in both APOCR and NDCG@n for the CurveNet backbone.
>
> | Method       | AOPCR   | NDCG@n  |
> |--------------|---------|---------|
> | PSM          | 1.371   | 0.218   |
> | CLAIM        | 1.363   | 0.252   |
> | **Additive**     | 0.615   | **0.266**  |
> | **Attention**     | 1.52    | 0.260 |
> | **Conjunctive**    | **2.66**    | 0.207   |
> | **Instance**    | 1.709 | 0.236   |
>
> We also show that PointMIL improves accuracy of CurveNet. To fairly demonstrate this, we present the original CurveNet architecture, the adapted architecture without MIL pooling, using the same classification head in the original paper, and then the adapted CurveNet with MIL pooling. The PointMIL version outperforms the original and adapted CurveNet on both real-world biological datasets by up to 3% mACC. Regarding ModelNet40, the PointMIL version of CurveNet outperforms the adapted version by 0.1%. However, the original CurveNet outperforms PointMIL by 0.3%.
>
> | Method                                | IntrA mACC (↑) | IntrA F1 (↑) | RBC mACC (↑) | RBC F1 (↑) | ModelNet40 mACC (↑) | ModelNet40 oACC (↑) |
> |---------------------------------------|----------------|--------------|--------------|------------|---------------------|---------------------|
> | CurveNet                              | 88.3           | 89.8         | 88.3         | 87.8       | -                   | 93.8                |
> | CurveNet$^{\dagger}$                  | 87.8           | 87.8         | 85.8         | 85.7       | 90.6                | 93.4                |
> | PointMIL (CurveNet$^{\dagger}$)       | 91.3 (+3.5)    | 89.9 (+2.1)  | 91.2 (+5.4)  | 90.5 (+4.8)| 91.0 (+0.4)         | 93.5 (+0.1)         |

---

> > ### Author Response · Authors · 2024-11-25
> > **Response to Reviewer yUX5 (Part 2)**
> >
> > ### **Misunderstanding regarding modification to backbone (ctd):**
> > 3. **General Applicability:** PointMIL is entirely backbone-agnostic and does not require any architectural changes to the backbone itself, unless there is some form of point sampling. No adaptations were made to PointNet and DGCNN. The only adaptations needed for other backbones is to remove point sampling. When we did this with CurveNet, we showed results on the original and adapted architectures to show that PointMIL outperforms both. The interpretability framework operates at the feature level, leveraging per-point features extracted by the backbone. This ensures that our method is generalisable across a wide range of architectures, including those without modifications.
> >
> > 4. **Alignment with Motivation:**
> > The motivation behind PointMIL remains fully intact: to provide a general, inherently interpretable method for point cloud classification. By demonstrating its integration with diverse backbones (e.g., PointNet, DGCNN, CurveNet, and Transformer-based), we illustrate its robustness and versatility without departing from the original goal.

---

> ### Author Response · Authors · 2024-11-25
> **Response to Reviewer yUX5 (Part 3)**
>
> ### **Novelty**
> >The novelty is limited: the authors' contributions primarily include (1) the feature extractor and (2) MIL pooling. However, the core idead of the feature extractor is merely a straightforward combination of existing Transformer and DGCNN approaches, lacking significant originality. In the MIL pooling, existing ideas are directly applied without any improvements tailored to the inherent characteristics of point cloud data. These factors restrict the paper's novelty, as it largely reflects a simple combination and adaptation of existing methods.
>
> Thank you for your thoughtful feedback. We appreciate the opportunity to clarify the novelty of our contributions and address these concerns:
>
> **Feature extractor:**
> While our feature extractor combines elements of existing Transformer and DGCNN approaches, the novelty lies in how it has been adapted to preserve per-point features and support inherently interpretable point cloud classification through MIL. For our feature extractor, we combine the strengths of Point Transformer and DGCNN to create a feature extractor that outputs high-quality point-level features to be used in MIL pooling. Specifically:
> * Unlike Point Transformer, which typically aggregate global features or reduce spatial resolution through point sampling and grouping, our approach retains per-point granularity throughout the network. This adaptation is crucial for enabling point-level interpretability, a core focus of our work.
> * Unlike DGCNN, our feature extractor includes contextual neighbourhood attention, which is explicitly designed to capture both local and global geometric relationships in point clouds. These additions are tailored to address the challenges of spatial irregularity and sparsity inherent in point cloud data.
>
>
> **MIL Pooling:**
> While we leverage existing MIL pooling mechanisms (e.g., Attention, Additive, Conjunctive, and Instance pooling), we go beyond simple application by:
> * **Introducing Contextual Attention:** This adaptation was specifically included for the use of attention-based MIL on point cloud data. This mechanism smooths importance weights using neighbourhood information, addressing sparsity and ensuring that point-level importance values reflect local geometric structures. This adaptation significantly enhances the interpretability and coherence of MIL pooling in the context of point clouds, as demonstrated in our experiments. Please see our ablation study on contextual attention in Figure 7.
>
> **Novelty in application:**
> Our contributions are not limited to the technical components themselves but also extend to their integration and application to point cloud data. To the best of our knowledge, this work represents the first inherently locally interpretable framework for point cloud classification. Existing point cloud methods typically rely on post-hoc approaches for interpretability, which lack transparency and direct integration into the prediction process. By combining MIL with point-specific feature extraction, we address this gap and demonstrate the feasibility of an inherently interpretable approach while improving performance of unchanged backbones.
>
> **Empirical Validation of Novelty:**
> The effectiveness of our innovations is reflected in the empirical results. Our method achieves state-of-the-art interpretability and classification metrics on real world biomedical datasets (IntrA and RBC) while maintaining competitive classification performance across benchmarks (ModelNet40, ShapeNetPart). These results validate the impact of our adaptations and their importance in enabling interpretability without compromising accuracy.

---

> ### Author Response · Authors · 2024-11-25
> **Response to Reviewer yUX5 (Part 4)**
>
> >The performance improvements are likely attributed mainly to the additional feature extraction network.
>
> To clarify, **there were no additional feature extraction networks added to any backbones.** Any improvement is due to MIL pooling alone which is a classification head.

---

> ### Author Response · Authors · 2024-11-25
> **Response to Reviewer yUX5 (Part 5)**
>
> ### **Questions:**
> >Although the authors provide visualizations of points deemed important for classification in the figures, these points are derived from the final pooling output. Why are these highlighted red points considered more important—just because they have higher importance scores? The current conclusions remain insufficiently convincing. Are there alternative methods that could offer supplementary validation? One possible approach could involve removing the red points and observing a significant accuracy drop, while removing the gray points leads to no substantial accuracy decrease. However, this is an intuitive idea and may not hold in actual experiments.
>
> We sincerely thank the reviewer for this insightful suggestion. To address the concern and provide supplementary validation, we have incorporated perturbation curves for each importance visualisation (Figures 2, 3, 5, and 6). These curves quantify the impact of removing points deemed most important by the model.
>
> To generate these perturbation curves, we followed this process:
>
> 1. We first computed the original logits for the predicted class using all points in the point cloud.
> 2. We sequentially removed points deemed most important (based on importance scores) in 5% intervals, recalculating the logits for the predicted class at each step. For example, we removed the top 51 points, recalculated the logits, then removed the top 102 points, and so on.
> 3. The resulting perturbation curves depict the decay in the predicted logit as more important points are removed.
>
> The results show a consistent decay in the model's predicted logit or class probability when the most important points are removed, validating that these points are indeed crucial to the model’s classification. For comparison, removing random or less important points does not lead to a substantial drop in logits, further reinforcing the reliability of the importance scores assigned by PointMIL.
>
> Additionally, we include qualitative examples where removing important points changes the model’s prediction. For instance:
> * Removing the most important points from a piano causes the model to misclassify it as a nightstand (Figure 5).
> * Similarly, removing important points from a chair results in a misclassification as a TV stand (Figure 5).
>
> Furthermore, AOPCR is a quantitative measure of exactly what the reviewer is suggesting. AOPCR measures the decrease in the model's logits when removing points deemed most important by the model compared to removing points randomly. Specifically:
> * The original logit for the predicted class is calculated with all points.
> * Points are removed sequentially in descending order of their importance scores, and the decrease in logits is recorded.
> * A parallel experiment removes the same number of randomly selected points, and the resulting decrease in logits is also recorded.
>
> A higher AOPCR indicates that removing important points causes a significantly greater decrease in logits than removing random points, quantitatively validating the relevance of these important points. Table 1 shows the AOPCR values for various methods and datasets, demonstrating that PointMIL consistently achieves higher AOPCR scores compared to baselines. This indicates that the points highlighted as important by PointMIL are indeed more critical for the model’s classification than random points.
>
> Similarly, NDCG@n is another quantitative measure that the points deemed important are actually discriminative of the class. NDCG@n compares the model's ranking of important points with a reference ranking based on their true contribution to the classification. We could measure NDCG@n on the IntrA dataset as we how ground truth annotations for the Aneurysm region of the blood vessel. Higher NDCG@n scores indicate that the model's importance scores align well with the actual impact of the points on its predictions. As seen in Table 1 In Table 1, PointMIL achieves consistently high NDCG@n scores across datasets and across backbones, further validating the quality of its point-level importance rankings.
>
> These analyses not only align with prior works on interpretability (Please see [1]) but also substantiate that the highlighted points significantly influence the model’s predictions. We hope these enhancements address the reviewer’s concerns and strengthen the validation of our method.
>
> [1] Joseph Early, et al. Inherently interpretable time series classification via multiple instance learning. In The Twelfth International
> Conference on Learning Representations, 2024

---

> > ### Comment · Reviewer_yUX5 · 2024-11-26
> >
> > Thank you for your effort and response. Your reply has partially addressed my previous concerns, such as the impact of additional parameters, which has led me to raise my score. However, I still have some concerns. For instance, the inclusion of CurveNet, as mentioned by another reviewer, originates from a 2021 work, raising questions about its adaptability to more contemporary architectures. Moreover, while you have introduced Contextual Attention within MIL pooling, your approach appears limited to attention-based MIL. For other existing architectures like PointNet and DGCNN, the methodology seems to involve a straightforward migration of prior work. Although such migration demonstrates some effectiveness, it does not strike me as the core innovative contribution of this work. Exploring more effective pooling mechanisms or developing MIL approaches specifically tailored for point cloud data could be an interesting research direction. Therefore, I raise my score to 5.

---

> > > ### Author Response · Authors · 2024-11-26
> > > **Exploring novel pooling**
> > >
> > > ### **Exploring novel pooling**
> > >
> > > We appreciate the reviewer’s suggestion to explore other novel pooling mechanisms tailored specifically for point cloud data. However, our current work focuses on integrating MIL pooling into point cloud classification to simultaneously offer interpretability and enhance classification performance. This is the **first inherently interpretable framework for point cloud classification**. While this is a simple amalgamation of numerous methods, this addresses a big gap in the literature, effectively.
> > >
> > > By applying MIL pooling across diverse backbones and diverse datasets of biological cells (IntrA and RBC) and everyday objects (ModelNet40) and demonstrating its effectiveness for quality interpretations and enhanced classification, we believe this work establishes a strong foundation for future research in this field. We envision this study as a stepping stone toward more advanced and specialised pooling mechanisms that further exploit the unique characteristics of 3D point cloud data. This recommendation has been explicitly highlighted in the conclusion of the updated manuscript as a potential avenue for further research.

---

> > > ### Author Response · Authors · 2024-12-01
> > > **Additional backbone (PointMLP)**
> > >
> > > We have added a third additional backbone to our analysis. As per the reviewers suggestion, we applied PointMIL to PointMLP. Specifically, we applied PointMIL to PointMLPElite by adapting the architecture not downsample the points. We showed the affect of this adaptation on classification results so that any difference in performance can then be attributed to the MIL pooling instead of this adaptation. We ran these experiments on the ModelNet40 dataset. The results are shown in Table 2 of the updated manuscript and in the table below:
> > > | Method                                  | mACC(↑) | oACC(↑) |
> > > |-----------------------------------------|---------|---------|
> > > | PointMLP            | **91.3** | **94.1** |
> > > | PointMLPElite                           | 90.9    | 93.6    |
> > > | PointMLPElite$^{\dagger}$               | 90.1    | 92.6    |
> > > | PointMIL (PointMLPElite$^{\dagger}$) | 90.5 (+0.4) | 93.5 (+0.9) |
> > >
> > > The results show how removing downsampling originally decays the accuracy. However, applying PointMIL to the adapted architecture improves accuracy back to near the original version. Importantly **this model is now inherently interpretable**.
> > >
> > > We further show how the PointMIL version of PointMLP has similar regions of interpretations to DGCNN, CurveNet, and Transformer on an example Bed and Plant from the ModelNet40 dataset. This is shown in Figure 6 of the updated manuscript.
> > >
> > > **We sincerely hope that this work has addressed all your initial concerns and we look forward to hearing your response.**

---

> > > ### Author Response · Authors · 2024-12-01
> > > **Request review of recent updates**
> > >
> > > Dear Reviewer yUX5,
> > >
> > > As the discussion period deadline is fast approaching (2nd December), we kindly wish to follow up regarding your feedback on our recent updates. To summarise, we have:
> > >
> > > * Completed experiments on additional newer backbones as suggested, including PointMLP and PointNeXt (in addition to CurveNet). This is 3 additional modern backbones during this rebuttal period. We showed that PointMIL improves classification performance on all backbones while offering inherent interpretability shown on two real world datasets of biological cells and common benchmark datasets suggested (ModelNet40 and ShapeNetPart). This showcases its adaptability to modern architectures.
> > > * Validated that the important points are indeed important through perturbation curves (as suggested).
> > > * Conducted the robustness to noise experiment showing that PointMIL improves robustness to noise on DGCNN (Figure 11, as suggested by Reviewer CZ3U). We also showed how interpretations are robust to noise (Figure 9).
> > > * Made all suggested clarifications.
> > > * Vastly increased and improved the discussion on interpretability.
> > >
> > >
> > > We truly appreciate your time and effort in reviewing our submission and look forward to hearing from you.
> > >
> > > Best regards, Authors

---

> ### Author Response · Authors · 2024-11-26
> **Addition of another backbone (PointNeXt, 2022)**
>
> We thank the reviewer for their constructive feedback and thoroughly appreciate them raising their score.
>
> ### **Adaptation to contemporary architectures**
>
> To address the concern regarding contemporary architectures, we have added **PointNeXt** [1] as an additional backbone to our experiments (as suggested by Reviewer CZ3U). PointNeXt is a more recent architecture based on PointNet++ [2]. This inclusion raises the total number of backbones evaluated to **five (PointNet, DGCNN, CurveNet, Transformer, and PointNeXt)** and the total number of models to **20**, showcasing a thorough and comprehensive analysis of MIL in interpretable point cloud classification.
>
> To adapt PointNeXt for PointMIL, we concatenated point-level features from the first layer of the encoder with the global features from the last encoder layer (repeated for every point), without modifying the architecture of PointNeXt itself. Thus obtaining point-level features forMIL pooling. This adaptation highlights the versatility of our approach in incorporating interpretability across different architectures.
>
> Again, we show that PointMIL provides higher quality interpretations that PSM and CLAIM as summarised below and now included in Table 1 of the revised manuscript:
>
> | Method         | AOPCR   | NDCG@n  |
> |----------------|---------|---------|
> | PSM           | 0.092   | 0.272   |
> | CLAIM         | 0.226   | 0.294   |
> | **Additive**        | 1.259   | 0.300   |
> | **Attention**         | 0.044   | 0.235   |
> | **Conjunctive**       | 1.531   | **0.310**   |
> | **Instance**         | **2.160**   | 0.285   |
>
> We also show that PointMIL improves the classification accuracy of PointNeXt across all datasets, as shown in the table below. Notably, PointMIL enhances **IntrA mACC by 2.8%** and **IntrA F1 by 1.5%**, further demonstrating its utility in improving interpretability and classification performance. While the improvements on the other datasets might be relatively smaller, PointMIL offers inherent interpretability into PointNeXt - a major benefit for adoption of these techniques in real-world scenarios.
>
> | Method                             | IntrA mACC ($\uparrow$)          | IntrA F1 ($\uparrow$)           |  RBC mACC ($\uparrow$)          | RBC F1 ($\uparrow$)           |  MN40 mACC ($\uparrow$)          | MN40 oACC ($\uparrow$)          |
> |------------------------------------|----------------------------|---------------------------|----------------------------|---------------------------|----------------------------|----------------------------|
> | PointNeXt                          | 91.8                       | 94.7                      | 86.1                       | 87.1                      | 90.8                       | 93.2                       |
> | **PointMIL** (PointNeXt)           | **94.6** (+2.8) | **96.2** (+1.5) | **87.6** (+1.5) | **88.2** (+0.4) | **90.5** (-0.3) | **93.3** (+0.1) |
>
> We have also included visualisations of PointNeXt interpretations with Additive and Conjunctive pooling in Figure 6 of the revised manuscript.
>
>
> While CurveNet was initially included as a baseline for comparison due to its strong performance and relevance, we acknowledge the importance of showcasing PointMIL on more contemporary architectures. **The addition of PointNeXt directly addresses this concern** and demonstrates that PointMIL can successfully integrate with and enhance more recent architectures.
>
>
>
> [1] Guocheng Qian, et al. Pointnext: Revisiting pointnet++ with improved training and scaling strategies. In S. Koyejo, S. Mohamed, A. Agarwal, D. Belgrave, K. Cho, and A. Oh (eds.), Advances in Neural Information Processing Systems, volume 35, pp. 23192–23204. Curran Associates, Inc., 2022.
>
> [2] Charles Ruizhongtai Qi, et al. Pointnet++: Deep hierarchical feature learning on point sets in a metric space. In I. Guyon, U. Von Luxburg, S. Bengio, H. Wallach, R. Fergus, S. Vishwanathan, and R. Garnett (eds.), Advances in Neural Information Processing Systems, volume 30. Curran Associates, Inc.,
> 2017.

---

> ### Author Response · Authors · 2024-12-02
> **Request reviewer yUX5 to review recent updates.**
>
> Dear Reviewer yUX5,
>
> As the discussion period deadline is fast approaching (end of day today), we kindly wish to follow up regarding your feedback on our recent updates. To summarise, we have:
>
> * Completed experiments on additional newer backbones as suggested, including PointMLP and PointNeXt (in addition to CurveNet). This is 3 additional modern backbones during this rebuttal period. We showed that PointMIL improves classification performance on all backbones while offering inherent interpretability shown on two real world datasets of biological cells and common benchmark datasets suggested (ModelNet40 and ShapeNetPart). This showcases its adaptability to modern architectures. Importantly, we discussed with Reviewer 1YBp how PointMLP and PointNeXt are the current state-of-the-art dedicated architectures for point-based classification without extra training data or pre-training etc.
> * Validated that the important points are indeed important through perturbation curves (as suggested).
> * Conducted the robustness to noise experiment showing that PointMIL improves robustness to noise on DGCNN (Figure 11, as suggested by Reviewer CZ3U). We also showed how interpretations are robust to noise (Figure 9).
> * Made all suggested clarifications.
> * Vastly increased and improved the discussion on interpretability.
> * **Now included additional experiments on the hardest version (PB_T50_RS) of ScanObjectNN (suggested by reviewer CZ3U and 1YBp).** This showed the practical value of PointMIL in real-world scenarios on a dataset with inherent complexities achieving comparable classification performance to the original backbones and offering **inherent interpretability**. Please see the table below:
>
> | Model                         | oACC         | mAcc          |
> |-------------------------------|--------------|---------------|
> | DGCNN (original)              | 78.1         | 73.6          |
> | DGCNN (PointNeXt training)    | 86.0 ± 0.5   |               |
> | PointMLPElite (original)      | 83.8 ± 0.6   | 81.8 ± 0.8    |
> | **PointMIL (DGCNN)**              | 85.6         | 83.0          |
> | **PointMIL (PointMLPElite)**      | 83.0         | 81.4          |
> | **PointMIL (Transformer)**        | 83.2         | 80.7          |
>
> We conducted these experiments to directly address the question of PointMIL's feasibility in real-world applications (Reviewer CZ3U), specifically under the challenging conditions of ScanObjectNN, which includes occlusions, noise, and background clutter. As shown in the table:
> - **PointMIL (DGCNN)** (trained with PointNeXt strategies) achieved an **overall accuracy (oACC) of 85.6%** and **mean accuracy (mAcc) of 83.0%**, which falls within the standard deviation range of the DGCNN (PointNeXt training) baseline.
> - **PointMIL (PointMLPElite)** also demonstrated strong performance, with PointMLP results comparable to or within the standard deviation range of their respective original baselines. This highlights PointMIL's adaptability to different backbone architectures.
> -  **PointMIL (Transformer)** (trained with PointNeXt strategies) demonstrated strong performance.
> - These results validate that PointMIL can maintain competitive classification accuracy on a real-world dataset with inherent complexities like those in ScanObjectNN further demonstrating practical value.
>
> Finally, the fact that **PointMIL transforms these models to become inherently interpretable is the major strength** for it's practical value in real-world scenarios where model interpretability is fundamental.
>
> We truly appreciate your time and effort in reviewing our submission and look forward to hearing from you.
>
> Best regards, Authors

---

> > ### Author Response · Authors · 2024-12-03
> > **Final request to reviewer yUX5 to review recent updates.**
> >
> > Dear Reviewer yUX5,
> >
> > Since your last response, we have made considerable updates which we kindly request that you review. To summarise, we have:
> >
> > * included experiments on an extra 2 backbones (PointMLP and PointNeXt) in addition to CurveNet. This brings the total number of backbones to 6 and the total number of newly proposed models to 24.
> > * included experiments on ScanObjectNN to showcase PointMIL on a dataset includes occlusions, noise, and background clutter.
> > * addressed concerns about novel pooling methods.
> >
> > We truly appreciate your time and effort in reviewing our submission and look forward to hearing from you.
> >
> > Best regards, Authors

---

### Official Review · Reviewer_1YBp · 2024-10-31

**Soundness:** 2
**Presentation:** 2
**Contribution:** 2
**Rating:** 5
**Confidence:** 4

**Summary:**

This paper introduces PointMIL, the first framework to apply MIL to point cloud classification. PointMIL provides fine-grained point-specific interpretability without post-hoc techniques.

**Strengths:**

1.	This paper is well-written, and the organization is great.
2.	The motivation is clear enough.

**Weaknesses:**

1.	The backbones and compared methods are limited. There are lots of methods that have been proposed for point cloud classification and segmentation. It is recommended to compare with them, including Point-MAE and PointTransformer V3.
2.	“Group features through k-nearest neighbours”, “Learned relative positional encoding”, and “Attention on the augmented features” are also widely used in point cloud processing, including PoinTr, Point-BERT, or DGCNN. It is recommended to move these parts into appendix and focus on your interpretation.
3.	The logic in L90-L94 is puzzling. “Post-hoc or inherently interpretable” and “local or global approaches” could be organized better.
4.	The citation in L100 should be (Tan & Kotthaus, 2022). The remaining part should also be carefully checked.

**Questions:**

1.	It is claimed that “most local interpretability methods for point cloud classification are post-hoc” and “no one has yet offered an inherently locally interpretable model for point cloud classification” seems conflict.
2.	Local and global features are widely studied in point cloud classification. Why your local approach is effective than others?
3.	MIL pooling has been widely used in 2D images. Please clarify the main difference between these methods. And it is also recommend to exploit the specific design for point cloud.
4.	The main contribution of this paper is the interpretation. However, the discussion about the interpretation is limited and the discussion about the network design has a great portion.
5.	The performance gain on ModelNet40 and ShapeNetPart is marginal. Moreover, more newly proposed backbones should be included as your backbone.

---

> ### Author Response · Authors · 2024-11-25
> **Response to Reviewer 1YBp (Part 1)**
>
> We want to thank the reviewer for their positive feedback and we appreciate the criticisms and advice to make our paper better. We address each weakness and question below:
>
> ### **Limited backbones**
> >The backbones and compared methods are limited. There are lots of methods that have been proposed for point cloud classification and segmentation. It is recommended to compare with them, including Point-MAE and PointTransformer V3.
>
> We appreciate this comment and agree that it is highly necessary for us to demonstrate PointMIL on more backbones. Therefore, to directly address this, we have extended our experiments to include an additional modern backbone, CurveNet [1], as discussed in Section 3.1. CurveNet is a widely adopted architecture that builds on recent advancements in capturing point cloud geometry through curve learning. We specifically chose CurveNet as the adaptations that were done to this method could be applied to any other method using farthest point sampling, including but not limited to PointMLP [2], and PointNeXt [3]. This resulted in four new models. Importantly, we adapted CurveNet to retain per-point features by removing point downsampling steps, ensuring compatibility with MIL pooling and enabling point-specific interpretability. This adaptation demonstrates the flexibility of PointMIL in integrating with modern architectures. We have included the interpretability results of CurveNet in Table 1, showing that PointMIL outperforms CLAIM and PSM in both APOCR and NDCG@n for the CurveNet backbone. Please see the table below:
>
> | Method       | AOPCR   | NDCG@n  |
> |--------------|---------|---------|
> | PSM          | 1.371   | 0.218   |
> | CLAIM        | 1.363   | 0.252   |
> | **Additive**     | 0.615   | **0.266**  |
> | **Attention**     | 1.52    | 0.260 |
> | **Conjunctive**    | **2.66**    | 0.207   |
> | **Instance**    | 1.709 | 0.236   |
>
>
> We also show that PointMIL improves accuracy of CurveNet. To fairly demonstrate this, we present the original CurveNet architecture, the adapted architecture without MIL pooling, using the same classification head in the original paper, and then the adapted CurveNet with MIL pooling. The PointMIL version outperforms the original and adapted CurveNet on both real-world biological datasets by up to 3% mACC. Regarding ModelNet40, the PointMIL version of CurveNet outperforms the adapted version by 0.1%. However, the original CurveNet outperforms PointMIL by 0.3%. The results on the three datasets are shown below. The adapted CurveNet architecture is shown with a $^{\dagger}$ .
>
> | Method                                | IntrA mACC (↑) | IntrA F1 (↑) | RBC mACC (↑) | RBC F1 (↑) | ModelNet40 mACC (↑) | ModelNet40 oACC (↑) |
> |---------------------------------------|----------------|--------------|--------------|------------|---------------------|---------------------|
> | CurveNet                              | 88.3           | 89.8         | 88.3         | 87.8       | -                   | 93.8                |
> | CurveNet$^{\dagger}$                  | 87.8           | 87.8         | 85.8         | 85.7       | 90.6                | 93.4                |
> | PointMIL (CurveNet$^{\dagger}$)       | 91.3 (+3.5)    | 89.9 (+2.1)  | 91.2 (+5.4)  | 90.5 (+4.8)| 91.0 (+0.4)         | 93.5 (+0.1)         |
>
> We chose CurveNet as the adaptations that were done to this method could be applied to any other method using farthest point sampling. We avoided PointTransformerv3 as this architecture is proposed for much larger scale point clouds and uses flash attention which require Ampere GPU’s which we do not have access to. Furthermore, Point-MAE is a self-supervised architecture that aims to learn a representation of a point cloud. Their classification experiments are based on transfer learning. PointMIL requires a supervised architecture and therefore Point-MAE is an unsuitable backbone.
>
> The only adaptation that we made to the original CurveNet architecture was to remove farthest point sampling. Other methods that include farthest point sampling such as PointMLP [2], PointNeXt [3], and Point Transformer v2/v3 could be similarly adapted to use PointMIL.
>
>
> [1] Tiange Xiang, et al. Walk in the cloud: Learning curves for point clouds shape analysis. In Proceedings of the IEEE/CVF International Conference on Computer Vision (ICCV), pp. 915–924, October 2021.
>
> [2]Xu Ma, Can Qin, Haoxuan You, Haoxi Ran, and Yun Fu. Rethinking network design and local geometry in point cloud: A simple residual MLP framework. In International Conference on Learning Representations, 2022.
>
> [3] Guocheng Qian, et al. Pointnext: Revisiting pointnet++ with improved training and scaling strategies. In S. Koyejo, S. Mohamed, A. Agarwal, D. Belgrave, K. Cho, and A. Oh (eds.), Advances in Neural Information Processing Systems, volume 35, pp. 23192–23204. Curran Associates, Inc., 2022.

---

> ### Author Response · Authors · 2024-11-25
> **Response to Reviewer 1YBp (Part 2)**
>
> ### **Focus on interpretability:**
> >“Group features through k-nearest neighbours”, “Learned relative positional encoding”, and “Attention on the augmented features” are also widely used in point cloud processing, including PoinTr, Point-BERT, or DGCNN. It is recommended to move these parts into appendix and focus on your interpretation.
>
> We appreciate this advice and have made reformatted our manuscript to focus on the interpretability as this is indeed the contribution of this paper. To summarise the main changes:
> * We have made included two new figures on interpretability.
> * We have included perturbation curves to every interpretation figure to show how the predicted logit decays when removing points deemed most informative by the model. This shows provides extra insight into the different interpretation methods. See Figures 2, 3, 5, and 6.
> * We have additionally demonstrated how the predictions for certain classes change to other classes when removing these important points. See Figure 5 where the classification of a piano changes to a night stand and a chair changes to a TV stand when removing the most important points.
> * We have visually shown how interpretations differ among backbones for the same representative shapes (Figure 6). Here, we saw that DGCNN, CurveNet, and Transformer had similar regions of importance when classifying a bed. These backbones focussed on the pillow, and backboard. However, PointNet focussed more on the mattress. Figure 6 also shows the perturbation curves for each pooling method and backbone on that shape.
> * We have included a discussion on the differences in the MIL pooling methods used including why they provide different levels of interpretability and when certain methods might be more beneficial over others. Please see lines 379-384 which state:
>
> "This superior performance of Additive and Conjunctive pooling can be attributed to their ability to better aggregate point-level importance scores. Additive pooling scales point features with their importance weights, preserving detailed information while focusing on relevant points. Conjunctive pooling further enhances this by independently computing attention weights and class-specific contributions, explicitly aligning the model's focus with the predicted class. In contrast, Instance pooling lacks this importance weighting, and Attention pooling does not offer class-specific explanations, which limits their interpretability."
>
> ### **Clarifying interpretability definitions**
> >The logic in L90-L94 is puzzling. “Post-hoc or inherently interpretable” and “local or global approaches” could be organized better.
>
> Thank you for pointing out the need for improved clarity in this section. We have revised the text to better organise the classification of interpretability methods for point clouds along two dimensions: the stage at which interpretability is introduced and the scope of the explanations provided. The revised text now reads:
>
> "Interpretability methods for point clouds can be classified along two key dimensions: (1) the stage at which interpretability is introduced and (2) the scope of the explanations provided. Regarding the stage, methods are either post-hoc or inherently interpretable. Post-hoc methods generate explanations after the model has made its predictions, often through additional analysis or approximation techniques. In contrast, inherently interpretable methods are designed to integrate interpretability into the model itself, producing explanations as part of the prediction process. Regarding the scope, methods are categorised as either local or global. Local approaches focus on explaining individual predictions, offering insights specific to a single input. Global approaches aim to provide a holistic understanding of the model's behaviour across all inputs."
>
> We hope this amendment addresses your concern and improves the clarity of this section. Please let us know if you believe that this should be amended further. Thank you for your helpful suggestion.
>
> ### **Citations:**
> >The citation in L100 should be (Tan & Kotthaus, 2022). The remaining part should also be carefully checked.
>
> We sincerely appreciate your close attention to our paper. This has been rectified. We have gone through all other citations to make sure that all are correct.

---

> ### Author Response · Authors · 2024-11-25
> **Response to Reviewer 1YBp (Part 3)**
>
> ### **Questions:**
>
> >It is claimed that “most local interpretability methods for point cloud classification are post-hoc” and “no one has yet offered an inherently locally interpretable model for point cloud classification” seems conflict.
>
> When we state that “most local interpretability methods for point cloud classification are post-hoc,” we mean that the majority of methods currently available provide local (point-level) explanations only after the model has made predictions (e.g., through saliency maps or perturbation-based techniques). These methods are designed for use with pre-trained models that are not inherently interpretable, meaning that the interpretations are not an output of the model.
>
> On the other hand, the statement “no one has yet offered an inherently locally interpretable model for point cloud classification” highlights a gap in the field: no prior work has presented a model that integrates local interpretability directly into the classification process. In other words, existing methods rely on post-processing for local explanations, while no inherently interpretable methods exist that provide such explanations as part of their forward pass.
>
> PointMIL provides local interpretations as a direct output of the model meaning that it is inherently locally interpretable. We hope that this clarifies this point. Please let us know if any further clarification is needed.

---

> ### Author Response · Authors · 2024-11-25
> **Response to Reviewer 1YBp (Part 4)**
>
> ### **Questions (ctd):**
> >Local and global features are widely studied in point cloud classification. Why your local approach is effective than others?
>
> Thank you for your thoughtful question. While global interpretation are highly useful, there is a gap in the literature of inherently interpretable local interpretations for point cloud classification. Furthermore, we focus on local interpretations because they offer several advantages over global interpretations, especially for point cloud classification:
>
> **Fine-Grained Explanations:** Local interpretations highlight the specific regions or points within a point cloud that contribute to the model's prediction. This level of granularity is essential for tasks where precise identification of critical features (e.g., a wing in an airplane model or a lesion in biomedical imaging) is required. In contrast, global interpretations provide a broader understanding of the model’s behaviour but often lack the resolution to pinpoint discriminative regions.
>
> **Relevance to Individual Predictions:** Local interpretations are particularly valuable for understanding individual predictions, as they reveal how specific features of a single input influenced the model’s decision. This contrasts with global interpretations, which summarise trends across the dataset and may not capture nuances relevant to specific cases.
>
> **Actionability in Real-World Applications:** In domains like biomedical imaging, local explanations can directly inform decisions by identifying key regions of interest, such as pathological structures. Global interpretations are less actionable in such cases, as they provide more abstract insights into the model's overall behaviour.
>
> Building on these strengths, our local approach is more effective than existing methods for several reasons:
>
> 1. **Inherently Interpretable Framework:**  Our method integrates local interpretability directly into the classification process using Multiple Instance Learning (MIL). This ensures that the importance of each point directly influences the model's predictions, making the interpretations more aligned with the decision-making process. Traditional post-hoc methods such as point saliency maps or class activation maps with CLAIM, on the other hand, are separate from the model and lack this direct connection.
>
> 2. **Enhanced with Contextual Attention:** We enhance local interpretability with contextual attention, which aggregates neighbourhood information to smooth importance weights and capture local geometric relationships. This allows our method to provide more coherent and meaningful explanations compared to methods that treat points independently. We show that this adaptation of attention-based MIL to point cloud analysis improves classification and interpretation performance.
>
> 3. **Class-Specific Explanations:** Our approach generates class-specific local explanations, distinguishing which regions contribute to specific predictions. This contrasts with many local methods that provide general relevance maps without focusing on specific class contributions.
>
> 4. **Generality Across Backbones:**
> Our local approach has been validated on multiple backbone architectures, including PointNet, DGCNN, CurveNet, and our transformer-based backbone, demonstrating its robustness and adaptability.
>
> 5. **Empirical Validation:** While CLAIM and PSM can be applied across backbones, PointMIL outperforms existing methods in interpretability metrics, as shown in Table 1 and Figures 2. For example, on the IntrA dataset, our method provides more precise and interpretable results for identifying critical regions compared to post-hoc local methods. Furthermore, CLAIM and PSM do not add to the classification performance of the model as these interpretability methods are done on a pre-trained model. PointMIL improved classification performance on all backbones across all datasets as shown in Table 2.
>
> For these reasons, we believe that we have shown that PointMIL is superior to other locally interpretable methods for point cloud classification.

---

> ### Author Response · Authors · 2024-11-25
> **Response to Reviewer 1YBp (Part 5)**
>
> ### **Questions (ctd):**
>
> >3. MIL pooling has been widely used in 2D images. Please clarify the main difference between these methods. And it is also recommend to exploit the specific design for point cloud.
>
> We appreciate that this might not have been clear in the original manuscript. To adapt MIL to the point cloud classification domain, we introduced a contextual attention discussed in Section 3.3. This mechanism specifically adapts MIL pooling to point cloud data by addressing the sparsity and fragmentation issues inherent to standard attention mechanisms. Below, we outline its contributions:
>
> * **Novel intuition:** This contextual attention modified the attention weights to reflect local geometric structures under the assumption that, in point cloud analysis and especially those in biological domain such as classifying healthy against diseased cells, regions of importance are generally close to one another. For example, if a point on an aneurysm is deemed important for classification, the surrounding points should be too.
>
> * **Challenges addressed:** MIL pooling inherently assumes independence among instances, which is unsuitable for point clouds. With attention-based MIL pooling, sparse attention distributions often fail to reflect the relationships among neighbouring points in point clouds. Contextual attention mitigates this by integrating neighbourhood-level information, producing smoother and more coherent attention maps.
>
> * **Performance improvements:** We showed that this contextual attention improved both classification performance (in terms of mACC, and F1) as well as interpretability performance (in terms of AOPCR and NDCG@n) through our ablation studies in Section 5.2.
>
> >4. The main contribution of this paper is the interpretation. However, the discussion about the interpretation is limited and the discussion about the network design has a great portion.
>
> We appreciate this advice and have reformatted our manuscript to focus on the interpretability as this is indeed the contribution of this paper. Please see our response to the similar weakness above. We hope that these amendments focus on the contribution of this paper which is interpretability.
>
> >5. The performance gain on ModelNet40 and ShapeNetPart is marginal. Moreover, more newly proposed backbones should be included as your backbone.
>
> While the performance gain on ModelNet40 and ShapeNetPart is marginal, on the real world datasets such as IntrA and RBC, PointMIL improved accuracies of the backbones by up to 11%. More importantly, the main contribution of PointMIL is to inherently include interpretations into a model. Furthermore, we have included another newly proposed backbone, CurveNet [1], to show how PointMIL can be used on methods that use farthest point sampling. Similar adjustments that we made to CurveNet can be apple to other architectures like PointMLP [2], and PointNeXt [3], ect.
>
> Please see our response to the weakness above discussing our analysis on the CurveNet backbone.
>
> 1] Tiange Xiang, et al. Walk in the cloud: Learning curves for point clouds shape analysis. In Proceedings of the IEEE/CVF International Conference on Computer Vision (ICCV), pp. 915–924, October 2021.
>
> [2]Xu Ma, Can Qin, Haoxuan You, Haoxi Ran, and Yun Fu. Rethinking network design and local geometry in point cloud: A simple residual MLP framework. In International Conference on Learning Representations, 2022.
>
> [3] Guocheng Qian, et al. Pointnext: Revisiting pointnet++ with improved training and scaling strategies. In S. Koyejo, S. Mohamed, A. Agarwal, D. Belgrave, K. Cho, and A. Oh (eds.), Advances in Neural Information Processing Systems, volume 35, pp. 23192–23204. Curran Associates, Inc., 2022.

---

> ### Comment · Reviewer_1YBp · 2024-11-26
> **Official Comment by Reviewer 1YBp**
>
> Thanks for answering all my concerns and providing a detailed response. Conducting experiments on CurveNet could enhance the overall quality of your manuscript. However, CurveNet is proposed in 2021, could you report more improvements on recently proposed methods? Moreover, ModeNet40, ScanObjectNN, and ShapeNet are widely used datasets. The performance gains in these datasets could demonstrate your superiority. Still, I appreciate the authors' commitment to improving the manuscript and providing experiments.

---

> > ### Author Response · Authors · 2024-11-26
> > **Another Additional Backbone (PointNeXt, 2022)**
> >
> > Thank you for your response. To showcase PointMIL further, we have **conducted another experiment on PointNeXt [1]**. PointNeXt is a more recent point-based architecture based on PointNet++ [2]. PointNeXt is another architecture that uses sampling and grouping. Therefore, in order to adapt PointNeXt for PointMIL such that point-level features were passed to the MIL pooling, we concatenated point-level features from the first layer in the encoder with the global features from the last layer of the encoder. Therefore, no adaptations were made to the architecture of this encoder, only the features used by the pooling. This is now discussed in the updated manuscript in Section 3.1. We have included the interpretability results of PointMIL with th PointNeXt backbone in Table 1, showing that **PointMIL outperforms CLAIM and PSM in both APOCR and NDCG@n for the PointNeXt backbone**. We have included the table below:
> >
> > | Method         | AOPCR   | NDCG@n  |
> > |----------------|---------|---------|
> > | PSM           | 0.092   | 0.272   |
> > | CLAIM         | 0.226   | 0.294   |
> > | **Additive**        | 1.259   | 0.300   |
> > | **Attention**         | 0.044   | 0.235   |
> > | **Conjunctive**       | 1.531   | **0.310**   |
> > | **Instance**         | **2.160**   | 0.285   |
> >
> > We have also included visualisations of PointNeXt interpretation with Additive and Conjunctive pooling in Figure ^ of the revised manuscript.
> >
> > We again show that PointMIL improves accuracy of PointNeXt. The PointMIL version outperforms the original PointNeXt on both real-world biological datasets by up to 2.8% mACC . The PointMIL version of PointNeXt outperforms the original version by 0.1% oACC. The results on the three datasets are shown below.
> >
> > | Method                             | IntrA mACC ($\uparrow$)          | IntrA F1 ($\uparrow$)           |  RBC mACC ($\uparrow$)          | RBC F1 ($\uparrow$)           |  MN40 mACC ($\uparrow$)          | MN40 oACC ($\uparrow$)          |
> > |------------------------------------|----------------------------|---------------------------|----------------------------|---------------------------|----------------------------|----------------------------|
> > | PointNeXt                          | 91.8                       | 94.7                      | 86.1                       | 87.1                      | 90.8                       | 93.2                       |
> > | **PointMIL** (PointNeXt)           | **94.6**+2.8 | **96.2** +1.5 | **87.6** +1.5 | **88.2** +0.4 | **90.5** -0.3 | **93.3** +0.1 |
> >
> > **We have now shown PointMIL on two newer and more advanced backbones bringing the total number of demonstrated backbones to 5 and the total number of different models to 20**. We sincerely hope that this proves the utility of PointMIL in adding inherent interpretability to point-based classifiers while improving classification accuracy.
> >
> > Finally, in our newly revised manuscript, we have moved “Group features through k-nearest neighbours”, “Learned relative positional encoding”, and “Attention on the augmented features” to the appendix and focussed even more on the interpretations, as per your suggestion in weakness 2:
> > >“Group features through k-nearest neighbours”, “Learned relative positional encoding”, and “Attention on the augmented features” are also widely used in point cloud processing, including PoinTr, Point-BERT, or DGCNN. It is recommended to move these parts into appendix and focus on your interpretation.
> >
> >
> > [1] Guocheng Qian, et al. Pointnext: Revisiting pointnet++ with improved training and scaling strategies. In S. Koyejo, S. Mohamed, A. Agarwal, D. Belgrave, K. Cho, and A. Oh (eds.), Advances in Neural Information Processing Systems, volume 35, pp. 23192–23204. Curran Associates, Inc., 2022.
> >
> > [2] Charles Ruizhongtai Qi, et al. Pointnet++: Deep hierarchical feature learning on point sets in a metric space. In I. Guyon, U. Von Luxburg, S. Bengio, H. Wallach, R. Fergus, S. Vishwanathan, and R. Garnett (eds.), Advances in Neural Information Processing Systems, volume 30. Curran Associates, Inc.,
> > 2017.

---

> > ### Author Response · Authors · 2024-12-01
> > **Request review of recent updates**
> >
> > Dear Reviewer 1YBp,
> >
> > As the discussion period deadline is fast approaching (2nd December), we kindly wish to follow up regarding your feedback on our recent updates. To summarise, we have:
> >
> > * Completed experiments on additional newer backbones as suggested, including PointMLP and PointNeXt (in addition to CurveNet). This is 3 additional modern backbones during this rebuttal period. We showed that PointMIL improves classification performance on all backbones while offering inherent interpretability shown on two real world datasets of biological cells and common benchmark datasets suggested (ModelNet40 and ShapeNetPart). This showcases its adaptability to modern architectures.
> > * Conducted the robustness to noise experiment showing that PointMIL improves robustness to noise on DGCNN (Figure 11, as suggested by Reviewer CZ3U). We also showed how interpretations are robust to noise (Figure 9).
> > * Made all suggested clarifications.
> > * Vastly increased and improved the discussion on interpretability.
> >
> > We truly appreciate your time and effort in reviewing our submission and look forward to hearing from you.
> >
> > Best regards, Authors

---

> > > ### Comment · Reviewer_1YBp · 2024-12-01
> > > **Response to Authors**
> > >
> > > I appreciate the authors' response to my questions. After reading the second rebuttal, I still have several concerns:
> > > 1. Please discuss the reason why PointMIL with PointNeXt undergoes performance decrease on MN40 mACC.
> > > 2. PointNeXt and PointMLP are both proposed in 2022. As I suggested in my first and second reviews, some up-to-date methods should be validated on your PointMIL. It is mainly due to most methods having achieved more than 94.0 MN40 oACC. Therefore, validating your method on these powerful methods could further prove your contributions.

---

> ### Author Response · Authors · 2024-12-01
> **Experiments on newer backbones (PointMLP)**
>
> We have added a third additional backbone to our analysis. As per the reviewers suggestion, we applied PointMIL to PointMLP. Specifically, we applied PointMIL to PointMLPElite by adapting the architecture not downsample the points. We showed the affect of this adaptation on classification results so that any difference in performance can then be attributed to the MIL pooling instead of this adaptation. We ran these experiments on the ModelNet40 dataset. The results are shown in Table 2 of the updated manuscript and in the table below:
> | Method                                  | mACC(↑) | oACC(↑) |
> |-----------------------------------------|---------|---------|
> | PointMLP            | **91.3** | **94.1** |
> | PointMLPElite                           | 90.9    | 93.6    |
> | PointMLPElite$^{\dagger}$               | 90.1    | 92.6    |
> | **PointMIL (PointMLPElite$^{\dagger}$)** | 90.5 (+0.4) | 93.5 (+0.9) |
>
> The results show how removing downsampling originally decays the accuracy. However, applying PointMIL to the adapted architecture improves accuracy back to near the original version. Importantly **this model is now inherently interpretable**.
>
> We further show how the PointMIL version of PointMLP has similar regions of interpretations to DGCNN, CurveNet, and Transformer on an example Bed and Plant from the ModelNet40 dataset. This is shown in Figure 6 of the updated manuscript.
>
> **We sincerely hope that this work has addressed all your initial concerns and we look forward to hearing your response.**

---

> ### Author Response · Authors · 2024-12-01
>
> Thank you for your timely follow-up.
>
> 1. The original PointNeXt paper [1] reports a ModelNet40 mACC of 90.8 ± 0.2% over three runs, while our experiment achieved a comparable 90.5%. This slight difference aligns with well known variability in results on ModelNet40 across different runs and implementations, as noted in the **CVMI-Lab/PAConv GitHub repository** [2]. ModelNet40 performance is known to fluctuate even under fixed seeds. Testing with pretrained models often ensures consistent results; however, PointMIL requires retraining, precluding direct use of pretrained weights. The randomness of ModelNet40 is also our motivation to test on real datasets of 3D biological cells (IntrA and RBC). Also, while state-of-the-art classification accuracy is an important goal, **PointMIL's primary contribution lies in introducing inherent interpretability without requiring post-hoc methods**. We have shown that PointMIL consistently improves interpretability metrics (e.g., AOPCR, NDCG@n) across multiple datasets and backbones, while maintaining competitive classification and **often improving** accuracy. The slight drop in ModelNet40 mACC with PointNeXt does not diminish the method's broader contributions.
>
>
> 2. While PointNeXt and PointMLP were both proposed in 2022, they are indeed the current state-of-the-art (on ModelNet40 and ShapeNetPart) in dedicated supervised learning (unlike Point-MAE) for point-based classification **without** additional training strategies listed below:
>    - **pre-training** (e.g., PointGPT (2024) [3], Mamba3D + Point-MAE (2024) [4],  PointView-GCN (2021) [5]).
>    - **fine-tuning** (e.g., PointGST (2024) [6]).
>    - **additional training data** e.g (ReCon (2023) [7]).
>    - **multi-modal** (e.g., ReCon++ (2024) [8]).
>
>
> The only end-to-end methods known to us to achieve over 94% oACC on ModelNet40 without these additional training strategies are PointMLP (94.5) and Mamba3D (94.1). And we have conducted analysis on the higher performing architecture, PointMLP. PointNeXt, CurveNet, and PointMLP are not outdated.
>
> Furthermore, achieving **inherently interpretable, point-level insights makes PointMIL a complementary addition** to these methods rather than a direct competitor.
>
>
> References:
>
>
> [1] Guocheng Qian, et al. *PointNeXt: Revisiting PointNet++ with Improved Training and Scaling Strategies*. Advances in Neural Information Processing Systems, 2022.
>
> [2] https://github.com/CVMI-Lab/PAConv/issues/9#issuecomment-873371422
>
> [3] Chen, G., et al. *PointGPT: Auto-regressively generative pre-training from point clouds*. Advances in Neural Information Processing Systems, 36. 2024
>
> [4] Xu Han, et al. *Mamba3D: Enhancing Local Features for 3D Point Cloud Analysis via State Space Model*. In Proceedings of the 32nd ACM International Conference on Multimedia (MM '24). 2024
>
> [5] S. S. Mohammadi, et al, "Pointview-GCN: 3D Shape Classification With Multi-View Point Clouds," IEEE International Conference on Image Processing (ICIP), 2021
>
> [6] Liang, D., et al. *Parameter-Efficient Fine-Tuning in Spectral Domain for Point Cloud Learning*. arXiv preprint arXiv:2410.08114, 2024.
>
> [7] Qi, Z., et al. *Contrast with Reconstruct: Contrastive 3D Representation Learning Guided by Generative Pretraining*. International Conference on Machine Learning (ICML), 2023.
>
> [8] Qi, Z., et al. *ShapeLLM: Universal 3D Object Understanding for Embodied Interaction*. European Conference on Computer Vision (ECCV), 2024.

---

> ### Author Response · Authors · 2024-12-02
> **Additional experiments on ScanObjectNN**
>
> ### **ScanObjectNN**
>
> >"Moreover, ModeNet40, ScanObjectNN, and ShapeNet are widely used datasets. The performance gains in these datasets could demonstrate your superiority."
>
> To address concerns,  we have conducted experiments on the hardest version of ScanObjectNN (PB_T50_RS) for DGCNN, PointMLPElite, and Transformer backbones. This concludes our analysis on all three suggested datasets. We have now shown PointMIL on a total of six datasets. Please see the table below:
>
> | Model                         | oACC         | mAcc          |
> |-------------------------------|--------------|---------------|
> | PointNet (original)              | 68.2         | 63.4          |
> | PointNet++ (original)              | 79.1         | 77.6          |
> | PointCNN (original)              | 78.5         | 75.1          |
> | DGCNN (original)              | 78.1         | 73.6          |
> | DGCNN (PointNeXt training)    | 86.0 ± 0.5   |               |
> | PointMLPElite (original)      | 83.8 ± 0.6   | 81.8 ± 0.8    |
> | **PointMIL (DGCNN)**              | 85.6         | 83.0          |
> | **PointMIL (PointMLPElite)**      | 83.0         | 81.4          |
> | **PointMIL (Transformer)**        | 83.2         | 80.7          |
>
> We conducted these experiments to directly address the question of PointMIL's feasibility in real-world applications, specifically under the challenging conditions of ScanObjectNN, which includes occlusions, noise, and background clutter. As shown in the table:
> - **PointMIL (DGCNN)** (trained with PointNeXt strategies) achieved an **overall accuracy (oACC) of 85.6%** and **mean accuracy (mAcc) of 83.0%**, which falls within the standard deviation range of the DGCNN (PointNeXt training) baseline.
> - **PointMIL (PointMLPElite)** also demonstrated strong performance, with PointMLP results comparable to or within the standard deviation range of their respective original baselines. This highlights PointMIL's adaptability to different backbone architectures.
> -  **PointMIL (Transformer)** (trained with PointNeXt strategies) demonstrated strong performance.
> - These results validate that PointMIL can maintain competitive classification accuracy on a real-world dataset with inherent complexities like those in ScanObjectNN further demonstrating practical value.
>
> Finally, the fact that **PointMIL transforms these models to become inherently interpretable is the major strength** for it's practical value in real-world scenarios where model interpretability is fundamental.

---

> ### Author Response · Authors · 2024-12-03
> **Request review of recent updates**
>
> Dear reviewer 1YBp,
>
> As the discussion period deadline is fast approaching, we kindly wish to follow up regarding your feedback on our recent most recent updates. To summarise, we have:
>
> * discussed the reason why PointMIL with PointNeXt undergoes slight performance decrease on MN40 mACC. This is due to the known variability in MN40 results discussed above.
> * discussed why PointMLP and PointNeXt are by no means out of date and are state-of-the-art end-to-end architectures that don't require additional training data, multiple modalities etc. also discussed above.
> * included experiments on ScanObjectNN to showcase PointMIL on a dataset includes occlusions, noise, and background clutter.
>
>
> We truly appreciate your time and effort in reviewing our submission and look forward to hearing from you.
>
> Best regards,
> Authors

---

### Official Review · Reviewer_9g12 · 2024-11-02

**Soundness:** 3
**Presentation:** 3
**Contribution:** 2
**Rating:** 6
**Confidence:** 3

**Summary:**

This paper introduces PointMIL, an inherently locally interpretable point cloud classifier using Multiple Instance Learning (MIL). It addresses a gap in existing classification methods, which either employ post-hoc interpretability techniques or focus on global interpretability. PointMIL comprises a feature encoder, implemented using either 3D transformers, DGCNN, or PointNet, to capture per-point features. A MIL pooling layer provides interpretability, with several types of MIL pooling methods explored, including Instance, Attention, Additive, and Conjunctive pooling.  A contextual prior is also injected into attention mechanism to learn local information. Experiments are conducted on medical dataset IntrA and general 3D object dataset ModelNet40.

**Strengths:**

1. The paper is well-written and easy to follow.
2. It claims to be the first work to achieve locally interpretable point cloud classification on a per-point basis.
3. The visual results are compelling, highlighting important regions for classification. Quantitative results on multiple classification benchmarks, such as IntrA and ModelNet40, and part segmentation benchmarks, including IntrA and ShapeNetPart, demonstrate performance improvements over baselines.

**Weaknesses:**

The technical contribution appears somewhat limited, as it primarily involves combining various typical point cloud encoders with existing MIL pooling methods for point cloud classification. I would have expected a deeper exploration of this combination. For example, were any modifications made to the MIL pooling methods to better adapt them to point cloud data? What specific challenges were encountered and addressed when applying MIL to this domain? Providing more insights into these aspects could help strengthen the contribution and highlight the distinctiveness of the approach.

Furthermore, in Tables 4, 5, and 6 in Appendix A.1, the paper presents varying performance across different MIL pooling methods. However, there is a lack of analysis explaining why certain methods perform better than others, and how these insights could guide the selection or adaptation of pooling methods in future work. Simply presenting the empirical results without such analysis misses an opportunity to make the work more insightful and inspiring.

The backbones used for evaluation are somewhat limited, as more recent and widely adopted models, such as Point Transformer (ICCV 2021) and sparse convolutional architectures, are not included. Evaluating PointMIL with these modern architectures would provide a more comprehensive comparison. I would appreciate it if the authors could discuss how PointMIL might be adapted to work with these newer backbones.  Additionally, a side-by-side comparison of the interpretability outputs for the same representative examples using different backbones would be valuable to determine if they highlight the same local regions of interest.

**Questions:**

1. In Lines 195-197, the paper claims to use five MIL pooling methods, but only four are listed: Instance, Attention, Additive, and Conjunctive. Could the authors please clarify whether a fifth pooling method was intended and omitted, or if this is a typo that should be corrected?
2.  In Table 2,  PointNet is reported to achieve 86.0 mACC, whereas the original paper cites 86.2 mACC. Could the authors please verify this discrepancy and explain? I wonder if this difference is due to variations in the implementation, experimental setup, or evaluation criteria.
3. Could the authors provide visual results for segmentation similar to Figures 2-5? Specifically, it would be interesting to include side-by-side comparisons of classification and segmentation interpretability outputs for a the same representative examples.

---

> ### Author Response · Authors · 2024-11-24
> **Response to Reviewer 9g12 (Part 1)**
>
> Firstly, we want to thank sincerely the reviewer for their positive feedback and for acknowledging the strengths of this paper.
>
> We address each weakness below:
>
> ### **Technical contribution:**
> >"The technical contribution appears somewhat limited, as it primarily involves combining various typical point cloud encoders with existing MIL pooling methods for point cloud classification. I would have expected a deeper exploration of this combination. For example, were any modifications made to the MIL pooling methods to better adapt them to point cloud data? What specific challenges were encountered and addressed when applying MIL to this domain? Providing more insights into these aspects could help strengthen the contribution and highlight the distinctiveness of the approach."
>
> We appreciate that this might not have been clear in the original manuscript. To adapt MIL to this domain, we introduced a contextual attention discussed in Section 3.3. This mechanism specifically adapts MIL pooling to point cloud data by addressing the sparsity and fragmentation issues inherent to standard attention mechanisms.
>
> Below, we outline its contributions:
> * **Novel intuition:** This contextual attention modified the attention weights to reflect local geometric structures under the assumption that, in point cloud analysis and especially those in biological domain such as classifying healthy against diseased cells, regions of importance are generally close to one another. For example, if a point on an aneurysm is deemed important for classification, the surrounding points should be too.
> * **Challenges addressed:** MIL pooling inherently assumes independence among instances, which is unsuitable for point clouds. With attention-based MIL pooling, sparse attention distributions often fail to reflect the relationships among neighbouring points in point clouds. Contextual attention mitigates this by integrating neighbourhood-level information, producing smoother and more coherent attention maps.
> * **Performance improvements:** We showed that this contextual attention improved both classification performance (in terms of mACC, and F1) as well as interpretability performance (in terms of AOPCR and NDCG@n) through our ablation studies in Section 5.2.
> We have now updated the manuscript to make this adaptation of MIL to point clouds clearer. Please see lines 61-65 and 72-74.
> Another important challenge was adapting the newer architectures to output point-level features. We have now included an additional newer backbone, CurveNet, and discuss this in detail below.
>
> ### **Discuss MIL pooling differences:**
> >Furthermore, in Tables 4, 5, and 6 in Appendix A.1, the paper presents varying performance across different MIL pooling methods. However, there is a lack of analysis explaining why certain methods perform better than others, and how these insights could guide the selection or adaptation of pooling methods in future work. Simply presenting the empirical results without such analysis misses an opportunity to make the work more insightful and inspiring.
>
> Thank you for your valuable feedback highlighting the need for an analysis of the varying performance across different MIL pooling methods. We agree that this is an essential aspect to explore. To address this, we have added a detailed comparative analysis in the revised manuscript in lines 378-384. We have also included the interpretability results for each pooling algorithm in the main text in Table 1. We have also shown perturbation curves for each figure showing how the logits of the original predicted class decay’s after removing the most important points. Please see Figures 2, 3, 5, and 6 in the new manuscript.
>  Below, we summarise the key insights:
>
> The varying performance across pooling methods stems from their mechanistic differences
> * **Attention pooling**  learns per-point weights, but it is not class-specific and rather a general measure of importance across classes. This means that attention-based pooling could assign high attention weights to points that are important but associated with a negative class (i.e., a class not being predicted). As a result, this method can sometimes fail to focus on truly discriminative regions for the target class, limiting its effectiveness in tasks requiring fine-grained, class-specific explanations.
> * **Instance pooling** classifies each point allowing class specific interpretations, however, this algorithm treats each point equally assuming that all points contribute equally to the final classification.
> * **Additive and Conjunctive pooling** combine attention weighting to specific points with point-level classifications allowing better class specific interpretations.
>
> We also showed that incorporating neighbourhood information through contextual attention significantly improved the interpretability and accuracy of attention-based pooling methods, highlighting its importance specifically in point cloud tasks.

---

> ### Author Response · Authors · 2024-11-24
> **Response to Reviewer 9g12 (Part 2)**
>
> **Insights for Future Work:** We provided recommendations for selecting pooling methods based on task-specific requirements. We have amended the manuscript to include:
>
> “The choice of pooling method should be guided by the specific requirements of the task and dataset characteristics. For tasks prioritising interpretability, Conjunctive pooling with contextual attention is recommended due to its class-specific focus. For applications prioritising simplicity, Instance pooling offers computational efficiency.”

---

> ### Author Response · Authors · 2024-11-25
> **Response to Reviewer 9g12 (Part 3)**
>
> ### **Additional backbones: (CurveNet)**
> > The backbones used for evaluation are somewhat limited, as more recent and widely adopted models, such as Point Transformer (ICCV 2021) and sparse convolutional architectures, are not included. Evaluating PointMIL with these modern architectures would provide a more comprehensive comparison. I would appreciate it if the authors could discuss how PointMIL might be adapted to work with these newer backbones.
>
> This comment was received by all reviewers and we completely agree with the need to demonstrate PointMIL on newer backbones. In response, we have extended our experiments to include an additional modern backbone, CurveNet [1], as discussed in Section 3.1. CurveNet is a widely adopted architecture that builds on recent advancements in capturing point cloud geometry through curve learning. We specifically chose CurveNet as the adaptations that were done to this method could be applied to any other method using farthest point sampling, including but not limited to PointMLP [2], and PointNeXt [3]. **This resulted in four new models**. Importantly, we adapted CurveNet to retain per-point features by removing point downsampling steps, ensuring compatibility with MIL pooling and enabling point-specific interpretability. This adaptation demonstrates the flexibility of PointMIL in integrating with modern architectures. We have included the interpretability results of CurveNet in Table 1, showing that PointMIL outperforms CLAIM and PSM in both APOCR and NDCG@n for the CurveNet backbone. Please see the table below:
>
> | Method       | AOPCR   | NDCG@n  |
> |--------------|---------|---------|
> | PSM          | 1.371   | 0.218   |
> | CLAIM        | 1.363   | 0.252   |
> | **Additive**     | 0.615   | **0.266**  |
> | **Attention**     | 1.52    | 0.260 |
> | **Conjunctive**    | **2.66**    | 0.207   |
> | **Instance**    | 1.709 | 0.236   |
>
>
> We also show that PointMIL improves accuracy of CurveNet. To fairly demonstrate this, we present the original CurveNet architecture, the adapted architecture without MIL pooling, using the same classification head in the original paper, and then the adapted CurveNet with MIL pooling. The PointMIL version outperforms the original and adapted CurveNet on both real-world biological datasets by up to 3% mACC. Regarding ModelNet40, the PointMIL version of CurveNet outperforms the adapted version by 0.1%. However, the original CurveNet outperforms PointMIL by 0.3%. The results on the three datasets are shown below. The adapted CurveNet architecture is shown with a $^{\dagger}$ .
>
> | Method                                | IntrA mACC (↑) | IntrA F1 (↑) | RBC mACC (↑) | RBC F1 (↑) | ModelNet40 mACC (↑) | ModelNet40 oACC (↑) |
> |---------------------------------------|----------------|--------------|--------------|------------|---------------------|---------------------|
> | CurveNet                              | 88.3           | 89.8         | 88.3         | 87.8       | -                   | 93.8                |
> | CurveNet$^{\dagger}$                  | 87.8           | 87.8         | 85.8         | 85.7       | 90.6                | 93.4                |
> | PointMIL (CurveNet$^{\dagger}$)       | 91.3 (+3.5)    | 89.9 (+2.1)  | 91.2 (+5.4)  | 90.5 (+4.8)| 91.0 (+0.4)         | 93.5 (+0.1)         |
>
>
> ### **Compare backbones visually:**
> >Additionally, a side-by-side comparison of the interpretability outputs for the same representative examples using different backbones would be valuable to determine if they highlight the same local regions of interest.
>
> We agree and have now shown this in the updated manuscript in Figure 5. DGCNN, CurveNet, and the Transformer backbone highlight similar regions of importance on a bed taken from the Modelnet40 dataset. We additionally show the perturbation curves for each MIL pooling algorithm on each backbone for this example.
>
> [1] Tiange Xiang, et al. Walk in the cloud: Learning curves for point clouds shape analysis. In Proceedings of the IEEE/CVF International Conference on Computer Vision (ICCV), pp. 915–924, October 2021.
>
> [2]Xu Ma, Can Qin, Haoxuan You, Haoxi Ran, and Yun Fu. Rethinking network design and local geometry in point cloud: A simple residual MLP framework. In International Conference on Learning Representations, 2022.
>
> [3] Guocheng Qian, et al. Pointnext: Revisiting pointnet++ with improved training and scaling strategies. In S. Koyejo, S. Mohamed, A. Agarwal, D. Belgrave, K. Cho, and A. Oh (eds.), Advances in Neural Information Processing Systems, volume 35, pp. 23192–23204. Curran Associates, Inc., 2022.

---

> > ### Author Response · Authors · 2024-11-26
> > **Addtional Modern Backbone (PointNeXt, 2022)**
> >
> > **In addition to our extension to CurveNet, we have now included PointNeXt [1} as a backbone for PointMIL**
> >
> > PointNeXt uses sampling and grouping, however, this architecture required minimal adaptation for integration with PointMIL, due to the first layer outputting point-level features. Specifically, we concatenated point-level features from the first encoder layer with the global features from the last encoder layer to enable MIL pooling without modifying the architecture itself. This approach demonstrates the adaptability of PointMIL to newer backbones while preserving architectural integrity.
> >
> > Our interpretability results, shown below, highlight that **PointMIL outperforms baseline methods such as CLAIM and PSM on both AOPCR and NDCG@n metrics for PointNeXt**, further validating its effectiveness. This is also shown in Table 1 of the revised manuscript.
> >
> > | Method         | AOPCR   | NDCG@n  |
> > |----------------|---------|---------|
> > | PSM            | 0.092   | 0.272   |
> > | CLAIM          | 0.226   | 0.294   |
> > | **Additive**       | 1.259   | 0.300   |
> > | **Attention**       | 0.044   | 0.235   |
> > | **Conjunctive**     | 1.531   | **0.310**   |
> > | **Instance**       | **2.160**   | 0.285   |
> >
> > We also evaluated PointMIL's impact on classification performance with PointNeXt. As shown below, **PointMIL improves PointNeXt's classification accuracy on real-world datasets like IntrA and RBC, achieving up to a 2.8% increase in mACC on IntrA.** On ModelNet40, PointMIL maintains competitive performance, with only a 0.3% decrease in mACC and 0.1% increase in oACC compared to the original PointNeXt. While the improvements on ModelNet40 are minor, it is important to point out that this is the first inherently interpretable framework for point cloud analysis. There often exists a trade-off between interpretation and classification, however **PointMIL offered interpretability without harming and often improving classification performance.**
> >
> > | Method                             | IntrA mACC (↑) | IntrA F1 (↑) | RBC mACC (↑) | RBC F1 (↑) | ModelNet40 mACC (↑) | ModelNet40 oACC (↑) |
> > |------------------------------------|----------------|--------------|--------------|------------|---------------------|---------------------|
> > | PointNeXt                          | 91.8           | 94.7         | 86.1         | 87.1       | 90.8                | 93.2                |
> > | **PointMIL (PointNeXt)**           | **94.6** (+2.8)| **96.2** (+1.5)| **87.6** (+1.5)| **88.2** (+0.4)| **90.5** (-0.3)     | **93.3** (+0.1)     |
> >
> > These results demonstrate the robustness and generalisability of PointMIL across diverse datasets, from biological data (IntrA and RBC) to everyday objects (ModelNet40). The inclusion of PointNeXt and CurveNet underscores the adaptability of PointMIL to contemporary architectures, highlighting its utility as the first inherently interpretable framework for point cloud classification. This work lays the foundation for future research in 3D point cloud interpretability, bridging the gap between performance and interpretability in this domain.
> >
> > [1] Guocheng Qian, et al. Pointnext: Revisiting pointnet++ with improved training and scaling strategies. In S. Koyejo, S. Mohamed, A. Agarwal, D. Belgrave, K. Cho, and A. Oh (eds.), Advances in Neural Information Processing Systems, volume 35, pp. 23192–23204. Curran Associates, Inc., 2022.

---

> ### Author Response · Authors · 2024-11-25
> **Response to Reviewer 9g12 (Part 4)**
>
> ### **Questions:**
> >1. In Lines 195-197, the paper claims to use five MIL pooling methods, but only four are listed: Instance, Attention, Additive, and Conjunctive. Could the authors please clarify whether a fifth pooling method was intended and omitted, or if this is a typo that should be corrected?
>
> We apologise for this misunderstanding. We were referring to the contextual pooling as the fifth pooling method. However, the contextual attention is an adaptation to all the attention-based pooling and, therefore, we have changed this to say “we evaluated four MIL pooling methods.”
>
> >2. In Table 2, PointNet is reported to achieve 86.0 mACC, whereas the original paper cites 86.2 mACC. Could the authors please verify this discrepancy and explain? I wonder if this difference is due to variations in the implementation, experimental setup, or evaluation criteria.
>
> We appreciate the close attention to detail. This was a typo. We have updated the table to show 86.2 mACC for PointNet.
>
> >3. Could the authors provide visual results for segmentation similar to Figures 2-5? Specifically, it would be interesting to include side-by-side comparisons of classification and segmentation interpretability outputs for a the same representative examples.
>
> We appreciate this comment and agree that this will improve the presentation of our paper. We have added segmentation results to our Appendix. Please see Figure 9 of the updated manuscript.

---

> ### Comment · Reviewer_9g12 · 2024-12-01
>
> Thank you for the clarification and the additional experiments. While they partly addressed my concerns, I still have the following reservations:
>
> - *Technical Contribution*: While I agree that the contextual attention mechanism is suitable for capturing local geometric structures in point cloud objects (as  in Figure 8), I am uncertain about the novelty of this approach. Specifically, averaging the attention of local points as an update seems incremental. Furthermore, as mentioned in Lines 474–498, although contextual pooling enhances classification and interpretability, it introduces a trade-off in computational efficiency due to increased time complexity. Thus, my concerns about the limited technical contribution remain.
>
> - *Visual Comparison*: In Figure 5, I could not find the suggested side-by-side comparisons of interpretability outputs for the same representative examples using different backbones. Similarly, side-by-side comparisons of classification and segmentation interpretability outputs for the same examples are also absent.

---

> > ### Author Response · Authors · 2024-12-01
> > **Response to Reviewer 9g12**
> >
> > Thank you for your response and taking the time to review our additional experiments.
> >
> > * **Technical contribution**: while our contextual attention is simple, we have demonstrated that this simple prior improves classification and interpretability. This does indeed carry a computational overhead, primarily due to the K-NN graph search. Future work could utilise k-d tree search to efficiently perform K-NN graph search. Other approaches could use radius graphs to speed up contextual attention.
> > * **Visual comparison**: this side-by-side comparison is shown in **Figure 6** of the finalised manuscript. Furthermore, we appreciate the suggestion for side-by-side interpretability outputs for segmentation and classification. However, for segmentation, the outputs are not interpretability outputs, but rather the segmentation predictions themselves. This segmentation analysis was primarily to showcase that while PointMIL is designed to add interpretability to point-based classifiers, it can also be used on segmentation algorithms without degrading performance.

---

> > ### Author Response · Authors · 2024-12-03
> > **Request Reviewer 9g12 to review final updates**
> >
> > Dear Reviewer 9g12,
> >
> > We kindly request that you review our recent updates. To summarise, we have:
> > * included experiments on ScanObjectNN.
> > * included PointMLP in addition to PointNeXt and CurveNet backbones during this rebuttal period. This means that we have demonstrated PointMIL on 6 backbones bringing the total number of inherently interpretable models proposed in this work to 24. And this was shown across 5 datasets.
> > * discussed our technical contribution further.
> > * pointed to the correct Figure depicting side-by-side comparisons of interpretability outputs for the same representative examples using different backbones.
> >
> > We sincerely thank you for your time in reviewing our work and look forward to hearing from you.
> >
> > Best regards,
> > Authors

---

### Author Response · Authors · 2024-11-25
**General repsponse to all reviewers**

We sincerely thank all reviewers for their thoughtful feedback and positive comments on our paper's novelty, innovativeness, organisation, and writing. We deeply appreciate the time and effort invested in providing detailed and constructive critiques. Each comment has contributed significantly to improving the quality and clarity of our work.

We have now addressed all weaknesses and questions raised by the reviewers. Changes in the manuscript are highlighted in blue. Below, we summarise the major updates:

### 1. **Additional backbone:**

In response to the unanimous recommendation to evaluate PointMIL on a more advanced backbone, we conducted experiments using  **CurveNet** [1]. This backbone originally employs sampling and grouping, which results in a loss of point-level features at the classification stage. We adapted CurveNet by removing sampling and grouping, allowing it to retain per-point features required for PointMIL’s interpretability framework. We therefore show how PointMIL can be used on a slightly adapted version of these architectures where the sampling and grouping is removed, allowing point-level features at the point of classification.
* The PointMIL version of CurveNet improved accuracy across all datasets and importantly provided inherent interpretability. Please see Tables 1, and 2, and Figure 5.
* Similar adaptations can be made to PointMLP, PointNeXt, Point Transformer, PointNet++, and any other backbone that uses sampling and grouping.

### 2. **Increased focus on interpretability:**

To emphasise our core contribution of inherent interpretability, we made the following enhancements:
* Added **perturbation curves** to all interpretability figures, illustrating how removing points deemed most important by the model leads to a significant drop in prediction confidence.
* Expanded discussions on the differences among MIL pooling methods and backbones, providing practical guidelines for selecting the appropriate methods based on task requirements.
* Included qualitative and quanttative examples demonstrating how interpretations vary across backbones for the same input, further validating the reliability of PointMIL’s interpretability outputs.


### 3. **Highlighting the Adaptation of MIL to Point Clouds:**

A major technical contribution of our paper is the adaptation of attention-based MIL to point cloud analysis through **contextual attention**. By assuming that points in close proximity should have similar attention weights, contextual attention smooths importance scores and reflects local geometric structures. This mechanism improved both interpretability and classification metrics, as evidenced by our experiments. We have now emphasised this contribution more clearly in the manuscript.

### 4. **Robustness Tests:**
To assess real-world applicability, we evaluated PointMIL’s robustness to noise on the IntrA dataset. PointMIL outperformed baselines like DGCNN, PointMLP, and PointNet under noisy conditions, showing its ability to focus on informative regions while ignoring noisy data.

### 5. **Real-World Applicability:**
PointMIL was evaluated on real-world biomedical datasets (IntrA, RBC), addressing practical use cases. While ScanObjectNN was beyond the scope due to time constraints, our robustness analysis and results on diverse datasets substantiate its applicability.


### 6. **Clarifications and Corrections:**
* Resolved typos and ambiguities (e.g., fifth pooling method clarification, corrected citations).
* Expanded explanations in the manuscript to address questions about downsampling impacts, MIL pooling advantages, and applicability to scene-level tasks.
* Clarified the distinction between post-hoc and inherent interpretability, outlining how PointMIL fills the gap in inherently interpretable local methods.

These improvements underscore PointMIL's novelty as the first inherently interpretable local framework for point cloud classification. We have now evaluated a total of 16 models on four datasets.

We hope that these revisions address the concerns and enhance the overall quality of our paper. Thank you again for the time and effort in reviewing our work.


[1] Tiange Xiang, et al. Walk in the cloud: Learning curves for point clouds shape analysis. In Proceedings of the IEEE/CVF International Conference on Computer Vision (ICCV), pp. 915–924, October 2021.

---

### Author Response · Authors · 2024-11-26
**Second revision (Extended PointMIL to PointNeXt)**

We thank the reviewers for their continued response. To further address the reviewers concerns, we have made a second revision, summarised below:

### **1. Second additional backbone (PointNeXt, 2022):**

After originally extending PointMIL to CurveNet, **we have now further extended PointMIL to PointNext [1]**, thus adding a second additional backbone during this rebuttal period. PointMIL offered high quality interpretations over CLAIM and PSM on PointNeXt, shown in the table below:
| Method         | AOPCR   | NDCG@n  |
|----------------|---------|---------|
| PSM            | 0.092   | 0.272   |
| CLAIM          | 0.226   | 0.294   |
| **Additive**       | 1.259   | 0.300   |
| **Attention**       | 0.044   | 0.235   |
| **Conjunctive**     | 1.531   | **0.310**   |
| **Instance**       | **2.160**   | 0.285   |

We also evaluated PointMIL's impact on classification performance with PointNeXt. As shown below, **PointMIL improves PointNeXt's classification accuracy on real-world datasets like IntrA and RBC, achieving up to a 2.8% increase in mACC on IntrA.** On ModelNet40, PointMIL maintains competitive performance, with only a 0.3% decrease in mACC and 0.1% increase in oACC compared to the original PointNeXt. While the improvements on ModelNet40 are minor, it is important to note that this is the first inherently interpretable framework for point cloud analysis. There often exists a trade-off between interpretation and classification, however **PointMIL offered interpretability without harming and often improving classification performance.**

| Method                             | IntrA mACC (↑) | IntrA F1 (↑) | RBC mACC (↑) | RBC F1 (↑) | ModelNet40 mACC (↑) | ModelNet40 oACC (↑) |
|------------------------------------|----------------|--------------|--------------|------------|---------------------|---------------------|
| PointNeXt                          | 91.8           | 94.7         | 86.1         | 87.1       | 90.8                | 93.2                |
| **PointMIL (PointNeXt)**           | **94.6** (+2.8)| **96.2** (+1.5)| **87.6** (+1.5)| **88.2** (+0.4)| **90.5** (-0.3)     | **93.3** (+0.1)     |

---

### **2. Further focus on interpretability:**
We have now further focussed our paper on interpretability. In addition to the first revision, we have increased the number of figures on interpretability to 7, and included lengthy discussions on the different MIL pooling methods and backbones. Therefore, we have restructured the manuscript according to the suggestion by Reviewer 1YBp:
> “Group features through k-nearest neighbours”, “Learned relative positional encoding”, and “Attention on the augmented features” are also widely used in point cloud processing, including PoinTr, Point-BERT, or DGCNN. It is recommended to move these parts into appendix and focus on your interpretation.

We have moved the feature extractor details to the Appendix (Appendix A).


**We hope that these second revisions address showcase PointMIL's versatility across modern backbones and import the quality of our work. Thank you again for the time and effort in reviewing our work. Please let us know if there are any more concerns.**


[1]]Guocheng Qian, et al. Pointnext: Revisiting pointnet++ with improved training and scaling strategies. In S. Koyejo, S. Mohamed, A. Agarwal, D. Belgrave, K. Cho, and A. Oh (eds.), Advances in Neural Information Processing Systems, volume 35, pp. 23192–23204. Curran Associates, Inc., 2022.

---

### Author Response · Authors · 2024-11-28
**Third revision (Extended to PointMLP (2022) and demonstrated robustness to noise)**

We thank the reviews again for their continued engagement and suggestions. We have uploaded a third revision to address the feedback. Here is a summary of the additional changes made in this revision:

### **PointMLP backbone**
We have included analysis of applying PointMIL to PointMLP [1] on ModelNet40. Specifically, we applied PointMIL to PointMLPElite by adapting the architecture to not downsample the points. We showed the affect of this adaptation on classification results so that any difference in performance can then be attributed to the MIL pooling instead of this adaptation. The results are shown in Table 2 of the updated manuscript and in the table below:
| Method                                  | mACC(↑) | oACC(↑) |
|-----------------------------------------|---------|---------|
| PointMLP [ma2022rethinking]             | **91.3** | **94.1** |
| PointMLPElite                           | 90.9    | 93.6    |
| PointMLPElite$^{\dagger}$               | 90.1    | 92.6    |
| \textsc{PointMIL} (PointMLPElite$^{\dagger}$) | 90.5 (+0.4) | 93.5 (+0.9) |

The results show how removing downsampling originally decays the accuracy. However, applying PointMIL to the adapted architecture improves accuracy back to near the original version. Importantly **this model is now inherently interpretable**.

We further show how the PointMIL version of PointMLP has similar regions of interpretations to DGCNN, CurveNet, and Transformer on an example Bed and Plant from the ModelNet40 dataset. This is shown in Figure 6 of the updated manuscript.


### **Robustness to noise**

Following the suggestion by Reviewer CZ3U, we have assessed PointMIL's robustness to noise. We initially performed this by comparing PointMIL with the Transformer backbone to other original architectures. However, Reviewer CZ3U pointed out that "previous work [2] has demonstrated that Transformers exhibit superior robustness when facing factors such as noise and rotation". Therefore, as per the suggestion, we performed those additional experiments by applying PointMIL to DGCNN and compared the robustness to noise with the original DGCNN on the IntrA dataset in terms of F1 score and mACC. This is now presented in Figure 11 in Appendix F of the updated manuscript. PointMIL outperformed the original DGCNN by substantial margins, demonstrating the robustness to noise.

To further address this concern, we have provided visualisations of interpretations of PointMIL using Transformer backbone on an Airplane from ModelNet40, shown in Figure 9 in the main of the updated manuscript. Figure 9 shows how, even when noisy
points are added to objects, PointMIL is still able to focus on salient 3D shape motifs, ignoring the noise.

All changes are shown in blue in the manuscript. **We hope that we have addressed all concerns with this revision. Please let us know about any concerns.**

[1] Xu Ma, et al. Rethinking network design and local geometry in point cloud: A simple residual MLP framework. In International Conference on Learning Representations, 2022

[2]  Sun, Jiachen, et al. "Benchmarking robustness of 3d point cloud recognition against common corruptions." arXiv preprint arXiv:2201.12296 (2022).

---

### Author Response · Authors · 2024-12-02
**Experiments on ScanObjectNN**

Reviewer CZ3U suggested experiments on ScanObjectNN as this dataset will showcase the practical value of PointMIL in real-world scenario's. To address this directly, we have conducted experiments on the hardest version of ScanObjectNN (PB_T50_RS) for DGCNN, PointMLPElite, and Transformer backbones. This concludes our analysis on all three suggested datasets. We have now shown PointMIL on a total of six datasets. Please see the table below:

| Model                         | oACC         | mAcc          |
|-------------------------------|--------------|---------------|
| PointNet (original)              | 68.2         | 63.4          |
| PointNet++ (original)              | 79.1         | 77.6          |
| PointCNN (original)              | 78.5         | 75.1          |
| DGCNN (original)              | 78.1         | 73.6          |
| DGCNN (PointNeXt training)    | 86.0 ± 0.5   |               |
| PointMLPElite (original)      | 83.8 ± 0.6   | 81.8 ± 0.8    |
| **PointMIL (DGCNN)**              | 85.6         | 83.0          |
| **PointMIL (PointMLPElite)**      | 83.0         | 81.4          |
| **PointMIL (Transformer)**        | 83.2         | 80.7          |

We conducted these experiments to directly address the question of PointMIL's feasibility in real-world applications, specifically under the challenging conditions of ScanObjectNN, which includes occlusions, noise, and background clutter. As shown in the table:
- **PointMIL (DGCNN)** (trained with PointNeXt strategies) achieved an **overall accuracy (oACC) of 85.6%** and **mean accuracy (mAcc) of 83.0%**, which falls within the standard deviation range of the DGCNN (PointNeXt training) baseline.
- **PointMIL (PointMLPElite)** also demonstrated strong performance, with PointMLP results comparable to or within the standard deviation range of their respective original baselines. This highlights PointMIL's adaptability to different backbone architectures.
-  **PointMIL (Transformer)** (trained with PointNeXt strategies) demonstrated strong performance.
- These results validate that PointMIL can maintain competitive classification accuracy on a real-world dataset with inherent complexities like those in ScanObjectNN further demonstrating practical value.

Finally, the fact that **PointMIL transforms these models to become inherently interpretable is the major strength** for it's practical value in real-world scenarios where model interpretability is fundamental.

---

### Author Response · Authors · 2024-12-03

We thank the reviewers for the efforts in reviewing our paper. We believe that we have thoroughly address all concerns raised by every reviewer. We summarise the updates during this rebuttal period below:
1. Experiments on 3 additional modern backbones (CurveNet, PointMLP, and PointNeXt) all showing improved accuracy inherent interpretability with PointMIL. This brings the total number of backbones to 6 and the total number of novel inherently interpretable models to 24.
2. Experiments on ScanObjectNN, demonstrating the practical utility of PointMIL on difficult datasets with noise and occlusion.
3. Validating interpretations with perturbation curves.
4. Substantial expansion on the interpretability sections of the paper.
5. Robustness-to-noise tests, proving that PointMIL increases a backbone's robustness to noise and that interpretations do not change under noisy conditions.
6. Highlighting our adaptation of MIL for point clouds.
7. Clarifications and corrections throughout the manuscript.

While our method is simple, we have rigorously demonstrated its effectiveness. We believe this represents a notable step forward in interpretable point cloud classification research.

---

### Note · Authors · 2025-02-18

I have read and agree with the venue's withdrawal policy on behalf of myself and my co-authors.

---

### Meta-Review · Area_Chair_bEiC · 2024-12-22

**Metareview:**

This paper focuses on integrating local interpretability directly into the point cloud classification process using Multiple Instance Learning.

It has received 4 reviews, with final ratings 6,6,5,5.  The authors provided extensive rebuttals, most notably, included additional experimental results with 3 additional modern backbones (CurveNet, PointMLP, and PointNeXt) and on ScanObjectNN.  They also successful at engaging several rounds of interactions with reviewers.  All reviewers appreciated authors' great efforts at rebuttals, and one reviewer (Reviewer 3) raised the final rating from 5 to 6.

However, despite all the clarification and additional results, Reviewers 1,2,4 still had major concerns.  Reviewer 1 found the technical contribution insignificant, without in-depth analysis or insights into the interpretability, no clear pattern out of different 3D backbones capturing various attentions for point cloud instances.   Reviewer 2 agreed with Reviewer 4 that the model's performance on ModelNet40 and ScanObjectNN was typically weaker than that of the original version, and questioned the solidity of experimental effectiveness.  Reviewer 4 questioned the practicality of this method based on weaker performance on ScanObjectNN, as it avoided downsampling which led to higher computational costs and reduced receptive fields.  Reviewer 4 also thought that this work directly applied multi-instance learning to 3D point clouds without making significant modifications specific to 3D data and considered the contribution insufficient for acceptance.  Based on all these lingering major concerns about the work, the AC recommends rejection.

**Additional Comments On Reviewer Discussion:**

Reviewers engaged in multiple rounds of interactions with the authors, and cross-referenced each other's comments.

---

### Decision · Program_Chairs · 2025-01-22

Reject